# EddyFormer: Accelerated Neural Simulations of Three-Dimensional Turbulence at Scale

**Yiheng Du**
UC Berkeley
yihengdu@berkeley.edu

**Aditi S. Krishnapriyan**
UC Berkeley, LBNL
aditik1@berkeley.edu

## Abstract

Computationally resolving turbulence remains a central challenge in fluid dynamics due to its multi-scale interactions. Fully resolving large-scale turbulence through direct numerical simulation (DNS) is computationally prohibitive, motivating data-driven machine learning alternatives. In this work, we propose *EddyFormer*, a Transformer-based spectral-element (SEM) architecture for large-scale turbulence simulation that combines the accuracy of spectral methods with the scalability of the attention mechanism. We introduce an *SEM tokenization* that decomposes the flow into grid-scale and subgrid-scale components, enabling capture of both local and global features. We create a new three-dimensional isotropic turbulence dataset and train EddyFormer to achieves DNS-level accuracy at $256^3$ resolution, providing a 30x speedup over DNS. When applied to unseen domains up to 4x larger than in training, EddyFormer preserves accuracy on physics-invariant metrics—energy spectra, correlation functions, and structure functions—showing domain generalization. On *The Well* benchmark suite of diverse turbulent flows, EddyFormer resolves cases where prior ML models fail to converge, accurately reproducing complex dynamics across a wide range of physical conditions. Code and project page are available at: `https://mrlazy1708.github.io/eddyformer`.

## 1 Introduction

Numerical simulation of fluid systems is essential for understanding and predicting complex flow phenomena [Van Dyke, 1982] in engineering and the natural sciences. Many critical applications, such as aerodynamics and weather forecast, rely on computational fluid dynamics (CFD) to simulate these flows. However, capturing flow patterns in turbulent flows [Pope, 2001] remains a fundamental challenge. In turbulence, *eddies*—coherent swirling structures—continuously interact and transfer energy across a wide range of length scales. The extent of these scales is determined by the Reynolds number, $Re$, a dimensionless quantity which represents the ratio of inertial to viscous forces. Fully resolving a turbulent flow requires $Re^{9/4}$ resolution, making DNS prohibitively expensive at high Reynolds number [Moin and Mahesh, 1998], even with supercomputers [Yeung et al., 2012, 2015].

Turbulence exhibits a characteristic cascade: energy is injected at large scales, transferred through the inertial range, and dissipated at the smallest scales where viscosity dominates. Statistical descriptions of turbulence assumes that the smallest scales become independent of the large-scale flow structures, which is known as the Kolmogorov's similarity hypothesis [Pope, 2001]:

> At sufficiently high Reynolds number, the statistics of small-scale motions have a universal form that is uniquely determined by viscosity and energy dissipation rate.

Based on this principle, large-eddy simulation [Smagorinsky, 1963] (LES) resolves only the large-scale structures while using theoretical models on smaller, subgrid scales (SGS). By filtering the governing equations, LES captures dominant flow features while relying on "closure" models to approximate unresolved dynamics at SGS. Despite its successes, LES faces challenges in wall-bounded and anisotropic turbulence [Piomelli, 1999], particularly in balancing accuracy and efficiency.

39th Conference on Neural Information Processing Systems (NeurIPS 2025).

Simulation time: 152 sec.          Simulation time: **4.86** sec.

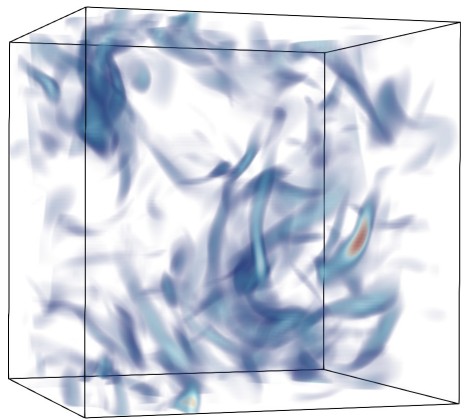
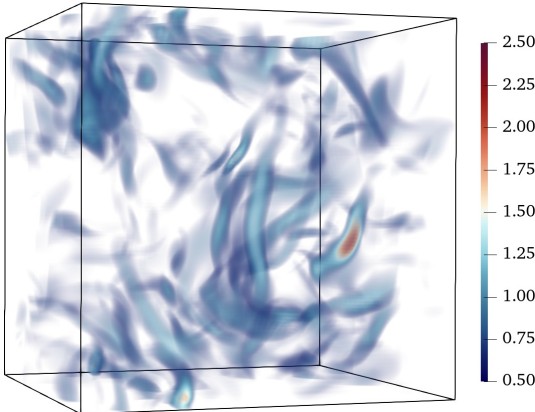

(a) $256^3$ numerical solution (16.3% L2 error).          (b) EddyFormer's prediction (18.2% L2 error).

Figure 1: Vorticity magnitude of forced turbulence with Reynolds no. $Re \approx 94$ at $t = 5$. EddyFormer reproduces the accuracy level and invariant flow statistics of a direct numerical simulation (DNS) at $256^3$ resolution, computed with highly efficient pseudo-spectral schemes [Canuto, 2007]. The reported L2 error is measured against a reference DNS at $384^3$ resolution. When benchmarked on a single NVIDIA A100 GPU, EddyFormer is 30× faster, achieving essentially real-time simulations.

Recent trends in ML have explored data-driven turbulence simulation [Li et al., 2020, Kochkov et al., 2021], with spectral-based neural network surrogates showing promising accuracy. Similar to spectral methods [Canuto, 2007], they approximate the solution using spectral bases and parameterize the dynamics through spectral transformations. For example, Fourier neural operators (FNOs) [Li et al., 2020] provide an expressive model for learning mappings on structured grids. However, scaling them to large problems with a wide range of fine-scale eddy dynamics remains challenging, as parameterizing large Fourier kernels is inefficient [Qin et al., 2024]. Likewise, Transformer architectures [Vaswani et al., 2017] face certain obstacles in fluid systems, as the quadratic growth of attention with mesh resolution may constrain their scalability in practice. These limitations suggest exploring architectures that balance efficiency with scalability to handle large-scale turbulence.

**Our contributions.**   We propose *EddyFormer*, a machine learning model for large-scale fluid flow simulation that captures multi-scale interactions through learned representations of the flow. The input is tokenized using the spectral/hp element method (SEM) [Karniadakis and Sherwin, 2005], which partitions the domain into coarse elements with spectral expansions in each subdomain. This tokenization combines the strengths of spectral modeling and scalability: local spectral expansions provide highly expressive representations, while each coarse element, i.e., a *SEM token*, yields a compact sequence that makes the attention mechanism efficient for capturing long-range interactions.

EddyFormer consists of two complementary streams: an *SGS stream* (§3.1), which captures localized eddy interactions; and an *LES stream* (§3.2), which models global, coherent structures. The LES stream is parameterized using multi-head attention applied to the filtered SEM tokens, significantly reducing computational complexity compared to self-attention on the full mesh. For the SGS stream, the Kolmogorov hypothesis suggests that small-scale dynamics can be modeled universally. We use spectral convolutions with a localized kernel to model the subgrid-scale dynamics.

EddyFormer demonstrates strong performance across a wide range of fluid flow problems, outperforming state-of-the-art ML models, as well as efficient numerical solvers at the same level of accuracy. For three-dimensional isotropic turbulence (§4.1, Fig. 1), EddyFormer achieves the accuracy of direct numerical simulation (DNS) at $256^3$ resolution while running 30× faster, reducing error by 30% compared to neural operator and Transformer baselines. When trained on two-dimensional turbulence (§4.2), it generalizes to domains up to four times larger than those seen in training by applying attention masking at test time, suggesting it learns domain-independent flow dynamics that can be adapted to general domain geometries. Benchmarks on other turbulent flows from The Well Ohana et al. [2025] (§4.3) show that EddyFormer accurately resolves turbulence driven by a wide range of physical conditions, including cases where other ML baselines fail to converge.

**Related works.** We briefly review ML approaches for turbulence here, with a more comprehensive survey of Transformer-based models and application domains presented in §A. Early attempts focused on learning data-driven turbulence closures for Reynolds-averaged Navier-Stokes (RANS)[Tracey et al., 2015, Ling et al., 2016] and large-eddy simulation (LES) [Beck et al., 2018, Font et al., 2021, List et al., 2022]. Wang et al. [2020] adopted the LES decomposition and modeled the mean and fluctuating fields separately. More recent works instead focus on direct simulation of turbulent flows using advanced neural networks architectures [Li et al., 2020, Tran et al., 2021, Wen et al., 2022, Kossaifi et al., 2023], showing promise but not yet consistently matching the accuracy of numerical solvers [Spalart, 2023, McGreivy and Hakim, 2024]. A promising alternative, "learned correction" [Kochkov et al., 2021, Shankar et al., 2025], uses low-accuracy solvers to improve accuracy. However, most previous works remain limited to small-scale two-dimensional flows.

## 2    Background

The Navier-Stokes (NS) equations [Kundu et al., 2024] describe the conservation of mass, momentum, and energy in fluid systems, capturing flows in a wide range of physical conditions. We focus on the incompressible flow described by the Boussinesq approximation for negligible density variations:

$$\frac{\partial \mathbf{u}}{\partial t} + \mathbf{u} \cdot \nabla \mathbf{u} = \nu \nabla^2 \mathbf{u} - \frac{1}{\rho_0} \nabla p,$$
$$\nabla \cdot \mathbf{u} = 0, \tag{1}$$

where $\mathbf{u}(t; x, y, z) \in \mathbb{R}^3$ is the velocity field on a domain $\Omega$, $\rho$ is the fluid density, $\nabla p$ is the pressure gradient, and $\nu$ is the kinematic viscosity. Given the initial velocity field $\mathbf{u}(0; \cdot)$ and certain boundary conditions on $\mathbf{u}$ and $p$, the task is to simulate the field $\mathbf{u}(t; \cdot)$ at a later time step $t$.

**Large-eddy simulation.** We adopt the methodology of large-eddy simulation (LES) in our model architecture. Classical LES aims to directly simulate the large-scale unsteady turbulent motions, while modeling the effects of small motions in theory. Given a (spatial) filter $G(\cdot, \cdot)$ on $\Omega$, the velocity field $\mathbf{u}(t; \cdot)$ is decomposed into its filtered field $\bar{\mathbf{u}}(t; \cdot) := G * \mathbf{u}(t; \cdot)$, and the corresponding subgrid-scale field (SGS) $\mathbf{u}' = \mathbf{u} - \bar{\mathbf{u}}$. The filtered velocity field $\bar{\mathbf{u}}$ is governed by the filtered momentum equation,

$$\frac{\partial \bar{\mathbf{u}}}{\partial t} + \bar{\mathbf{u}} \cdot \nabla \bar{\mathbf{u}} = \nu \nabla^2 \bar{\mathbf{u}} - \nabla \cdot \tau - \frac{1}{\rho_0} \nabla \bar{p}, \tag{2}$$

where $\tau_{ij} := \overline{u_i u_j} - \bar{u}_i \bar{u}_j$ is the residual stress (or SGS) tensor. This filtered equation is unclosed, in terms of the SGS tensor being approximated based on the filter $G$ and the filtered field $\bar{\mathbf{u}}$. Many classical turbulence models express $\tau$ as an eddy viscosity $\tau_{ij} \approx -2\nu_t \bar{S}_{ij}$, where $\bar{S}$ is the resolved strain rate tensor and $\nu_t$ is the eddy viscosity to be modeled by the corresponding SGS model.

**Spectral element method.** We use the standard spectral element method (SEM) to parameterize the solution field $\mathbf{u}(t; \cdot)$. The physical domain $\Omega$ is discretized into $H$ structured elements $\Omega = \cup_{h \leq H} \Omega_h$, where each element $\Omega_h$ is mapped to an unit cube through a geometric transformation. We assume a uniform mesh with cell size $\Delta$ and denote the local coordinates on $\Omega_h$ as $0 \leq \xi_h^x, \xi_h^y, \xi_h^z \leq 1$.

The local field $\mathbf{u}_h(t; \xi_h)$ on $\Omega_h$ is expanded using higher-order bases. Let $\{\phi_m\}_m$ be a set of one-dimensional basis on $[0, 1]$, their full tensor product forms a multi-dimensional basis set $\Phi_p(\xi) := \Pi_i \phi_{p_i}(\xi_i)$. The local field $\mathbf{u}_h(t; \cdot)$ is then approximated as,

$$\mathbf{u}_h(t; \xi_h) = \sum_{|p| \leq M} \tilde{\mathbf{u}}_{hp}^t \Phi_p(\xi_h), \tag{3}$$

where $\tilde{\mathbf{u}}_h^t \in \mathbb{R}^{3M^3}$ are the expansion coefficients of the $h$'th element. In contrast to spectral methods, the standard spectral basis must be modified to enforce a certain order of continuity across the element boundaries. In this work, we consider $C^0$-continuous boundary-interior decomposition. Given a set of orthogonal polynomials[1] $\{p_m\}_m$, their corresponding modal $p$-type basis is defined as,

$$\phi_m(\xi) = \begin{cases} 1 - \xi & m = 0, \\ (\xi - \xi^2) p_{m-1}(\xi) & 0 < m < M, \\ \xi & m = M, \end{cases} \tag{4}$$

where $\phi_0$ and $\phi_M$ are the shared left and right boundary modes, respectively.

---

[1] In this work, we implement the (shifted) Chebyshev polynomials $T_m(x) = \cos(m \arccos(2x - 1))$, and the Legendre polynomials $P_m(x) = \frac{1}{2^m m!} \frac{d^m}{dx^m}(x^2 - 2x)^m$, as the orthogonal polynomial set $p_m$.

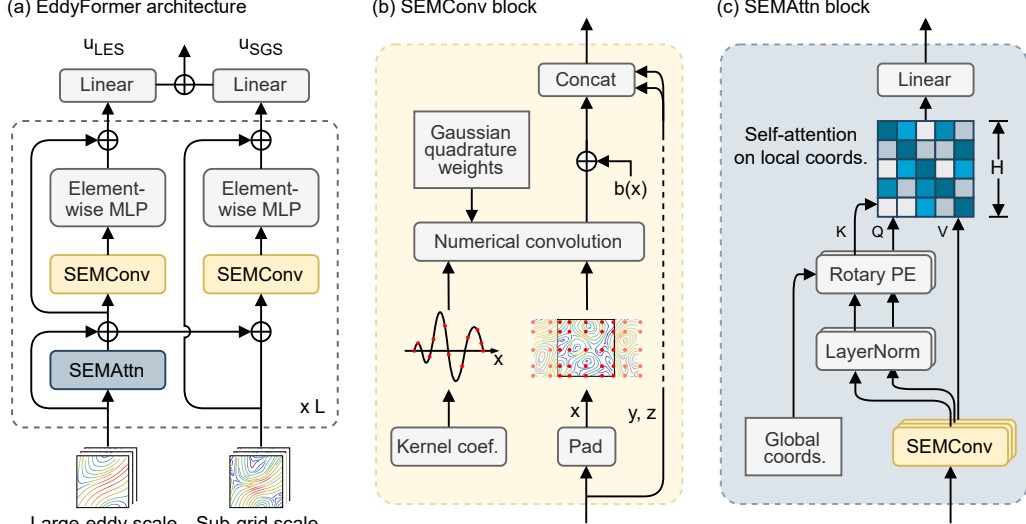

Figure 2: Our proposed architecture: *EddyFormer*. **(a)** The PDE initial conditions are interpolated using spectral element methods (SEM). The large-eddy simulation (LES) field and subgrid-scale (SGS) field are transformed through the LES stream (§3.2) and SGS stream (§3.1), respectively. The SGS stream consists of **(b)** SEM-based convolution blocks to model local eddy dynamics, while the LES stream consists of **(c)** SEM-based attention blocks to capture long-range dependencies.

## 3 Model design choices for EddyFormer

We introduce the EddyFormer architecture (Fig. 2). EddyFormer parameterizes the solution field using SEM bases defined in Eqn. (3). For a fixed domain discretization $\Omega = \cup_h \Omega_h$ with $H = N^3$ elements and $P = M^3$ modes per element, EddyFormer maps the spectral representation of the initial condition $\mathbf{u}_0$ to the prediction $\mathbf{u}(t; \cdot)$ at a fixed time $t$, based on a chosen SEM basis $\Phi$:

$$\mathbf{u}(0; ) = \sum_{|p| \leq M} \tilde{\mathbf{u}}^0_{hp} \Phi_p \mapsto \sum_{|p| \leq M} \tilde{\mathbf{u}}^t_{hp} \Phi_p = \mathbf{u}(t; \cdot) \quad \text{on } \Omega_h, \tag{5}$$

where $\tilde{\mathbf{u}}^t \in \mathbb{R}^{3N^3 M^3}$ is the output of the model.

EddyFormer decomposes the output as the sum of an LES-filtered part $\mathbf{u}_{\mathsf{LES}}$ and the residual field $\mathbf{u}_{\mathsf{SGS}}$. The residual coefficients $\tilde{\mathbf{u}}_{\mathsf{SGS}}$ are unfiltered and parameterized by an *SGS stream* (§3.1). For the LES filter, we simply use a spectral cutoff filter with $k_{\mathsf{max}} < M$ modes on each axis, parameterizing the LES-filtered coefficients $\tilde{\mathbf{u}}_{\mathsf{LES}} \in \mathbb{R}^{3N^3 k_{\mathsf{max}}^3}$ by an *LES stream* (§3.2). In §C.2.1, we perform an ablation study on the LES-SGS splitting with different $k_{\mathsf{max}}$ to verify its effectiveness.

$$\begin{array}{l} \text{SGS stream:} \quad \mathbf{u}^{(0)} \to \mathbf{u}^{(1)} \to \ldots \to \mathbf{u}^{(L)} \to \mathbf{u}_{\mathsf{SGS}} \\ \text{LES stream:} \quad \bar{\mathbf{u}}^{(0)} \to \bar{\mathbf{u}}^{(1)} \to \ldots \to \bar{\mathbf{u}}^{(L)} \to \mathbf{u}_{\mathsf{LES}} \end{array} \Bigg\} \longrightarrow \mathbf{u} = \mathbf{u}_{\mathsf{LES}} + \mathbf{u}_{\mathsf{SGS}}. \tag{6}$$

### 3.1 SGS stream for local motions

The input is first interpolated onto the chosen SEM basis $\Phi$, and then projected to the initial feature, $\mathbf{u}^{(0)} = W_{\mathsf{in}} \mathbf{u}(0; \cdot)$, and the output is the projection of the final feature, $\mathbf{u}_{\mathsf{SGS}} = W_{\mathsf{SGS}} \mathbf{u}^{(L)}$. In each layer $l$, the locals fields $\mathbf{u}_h^{(l)}$ receive information from the LES stream and then perform convolutions:

$$\mathbf{u}_h^{(l+1)} = \mathbf{u}_h^{(l)} + \mathsf{FFN}(\mathsf{SEMConv}_h(\mathbf{u}_h^{(l)} + \epsilon\, \bar{\mathbf{u}}_h^{(l)})), \tag{7}$$

where FFN is a two-layer feed-forward network applied on the collocation points independently, and $\epsilon$ is a learnable scalar. SEMConv is our SEM-based convolution block, as introduced below.

**SEM-based convolution.** We extend the powerful spectral convolution [Li et al., 2020] as the basic block in our architecture. The SEMConv block parameterizes convolution in SEM bases. For a given

field $\mathbf{u}$ with $d$ components, we parameterize its convolution with a compact kernel $\mathbf{k}$ that is supported on a window of size $s$,

$$\mathsf{SEMConv}(\mathbf{u})(\mathbf{x}') = \int_{|\mathbf{x}| \leq s/2} \mathbf{k}(\mathbf{x})\mathbf{u}(\mathbf{x}' - \mathbf{x})^T d\mathbf{x}. \qquad (8)$$

Note that $s > 0$ is a hyperparameter to be determined. We found that a small value of $s$ (compared to the domain length-scale) is usually expressive enough. Due to the success of Fourier factorization techniques [Tran et al., 2021], we also restrict the convolution kernel to axial form,

$$\mathbf{k}(x, y, z) = \sum_m \tilde{\mathbf{k}}_m [e^{i\pi mx/s} \ e^{i\pi my/s} \ e^{i\pi mz/s}]^T, \qquad (9)$$

where $\tilde{\mathbf{k}}_m \in \mathbb{C}^{d \times d \times 3}$ are the learnable Fourier coefficients. The convolution is computed by directly evaluating Eqn. (8) using numerical quadrature rules (see Algorithm 1). Since the number of modes per element is small compared to the total number of modes, the computational cost of SEMConv remains comparable to that of FFT-based convolution with global kernels (see Tab. 1 for comparison).

**Inclusion of LES features.** The inclusion of LES features in the SGS stream, Eqn. (7), is justified by the turbulence cascade. Large eddies transport the turbulent kinetic energy downwards, while smaller eddies primarily dissipate that energy. Classical closures—such as Smagorinsky [1963] and Spalart and Allmaras [1992]—reflect this hierarchy by parameterizing subgrid stress $\tau$ in Eqn. (2) using the resolved large-scale flow field. Similarly, we also find that the inclusion of LES features into the SGS features is crucial for EddyFormer's performance, but not vice versa.

## 3.2 LES stream for global motions

The initial LES feature $\bar{\mathbf{u}}^{(0)}$ is spectrally truncated to $k_{\mathsf{max}}$ modes, and the output is the projection of the final feature, $\mathbf{u}_{\mathsf{LES}} = W_{\mathsf{LES}}\bar{\mathbf{u}}^{(L)}$. The local fields $\bar{\mathbf{u}}_h^{(l)}$ are regarded as *SEM tokens*. In each layer $l$, the SEM tokens are globally mixed with other tokens first, and then locally transformed. Specifically,

$$\begin{aligned} \bar{\mathbf{u}}_h^{(l)} &\leftarrow \bar{\mathbf{u}}_h^{(l)} + \mathsf{SEMAttn}_h(\bar{\mathbf{u}}^{(l)}), \\ \bar{\mathbf{u}}_h^{(l+1)} &= \bar{\mathbf{u}}_h^{(l)} + \mathsf{FFN}(\mathsf{SEMConv}_h(\bar{\mathbf{u}}^{(l)})), \end{aligned} \qquad (10)$$

where SEMAttn is our SEM-based self-attention block, which we introduce below.

**SEM-based attention.** We utilize the attention mechanism to model interactions between SEM tokens. In SEMAttn, the SEM tokens $\bar{\mathbf{u}}_h^{(l)}$ are mixed based on their *local* coordinates. Specifically, these features are projected using convolutions, $\bar{\mathbf{u}}^{\mathsf{Q/K/V}} = \mathsf{SEMConv}^{\mathsf{Q/K/V}}(\bar{\mathbf{u}}^{(l)})$, and the SEMAttn is defined by mixing all features on the same *local* coordinate $\xi$ using self-attention mechanism,

$$\mathsf{SEMAttn}_h(\bar{\mathbf{u}}^{(l)})(\xi) = \sum_{h' \leq H} \mathsf{SoftMax}_{h'}(\bar{\mathbf{u}}_h^{\mathsf{Q}}(\xi) \cdot \bar{\mathbf{u}}_{h'}^{\mathsf{K}}(\xi)) \, \bar{\mathbf{u}}_{h'}^{\mathsf{V}}(\xi), \qquad (11)$$

which is implemented by performing self-attention on each collocation point independently. Note that SEMAttn strictly preserves the continuity of the input features, regardless of the discretization scheme of SEM representation. Similar to standard Transformers, we also add learnable bias terms, apply layer normalization[2], and perform position encoding to the key and query features.

In contrast to mesh-based Transformers, the sequence length, or the number of SEM tokens, equals the number of spectral elements $N^3$, which is orders of magnitude smaller than the number of resolved modes $N^3M^3$. This avoids the unfavorable quadratic scaling of standard attention mechanism.

**Rotary position encoding.** While vision Transformers [Dosovitskiy et al., 2020] are insensitive to position encoding, we find that turbulence simulation relies on substantial positional information in SEMAttn. Inspired by Su et al. [2024], we rotate the key and query fields based on their *global* coordinates, namely, for the $k$'th channel, the rotation along $x$ axis is defined by,

$$\begin{aligned} \bar{\mathbf{u}}_{h,2k}^{\mathsf{Q/K}}(x, y, z) &\leftarrow \cos(kx)\bar{\mathbf{u}}_{h,2k}^{\mathsf{Q/K}}(x, y, z) - \sin(kx)\bar{\mathbf{u}}_{h,2k+1}^{\mathsf{Q/K}}(x, y, z), \\ \bar{\mathbf{u}}_{h,2k+1}^{\mathsf{Q/K}}(x, y, z) &\leftarrow \sin(kx)\bar{\mathbf{u}}_{h,2k}^{\mathsf{Q/K}}(x, y, z) + \cos(kx)\bar{\mathbf{u}}_{h,2k+1}^{\mathsf{Q/K}}(x, y, z). \end{aligned} \qquad (12)$$

The rotation along $y$ and $z$ axes are similar, and the final results are concatenated together. Due to the properties of rotation, it's apparent that the SEMAttn layer is also strictly translational invariant.

---

[2]We don't apply normalization on any other hidden features, since it was found to hurt model performance. In order to regularize gradients, we initialize all convolution kernels with a random variable of magnitude $10^{-7}$.

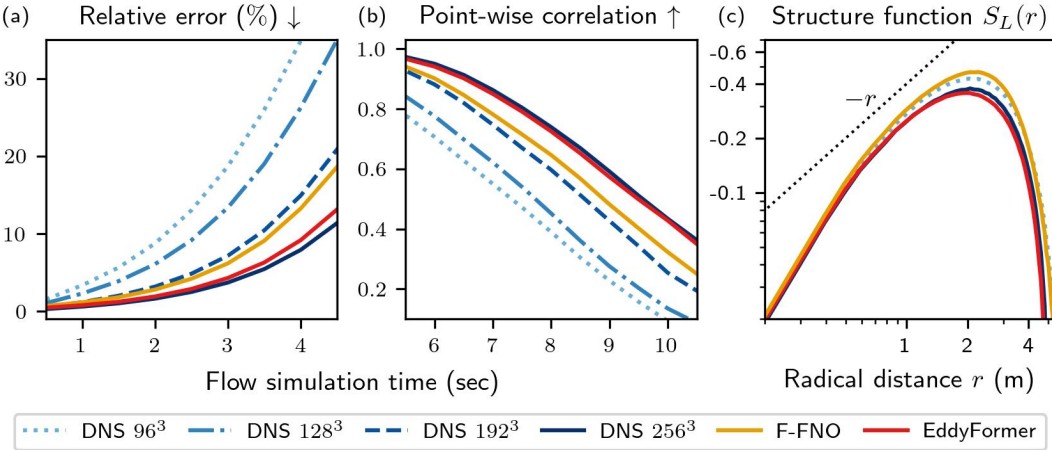

Figure 3: Performance of models and DNS on `Re94` (§4.1), a 3D homogeneous isotropic turbulence fluid flow. The models learn to correct DNS at $96^3$ resolution, and we report evaluation metrics relative to a reference DNS at $384^3$ resolution. EddyFormer achieves the accuracy of DNS at $256^3$ resolution in terms of **(a)** relative L2 error for $t \leq 5$ and **(b)** correlation for $5 \leq t \leq 10$ which measures field similarities; for **(c)** longer prediction rollouts up to 20 seconds, EddyFormer also captures the third-order structure function, Eqn. (18), which measures the velocity skewness and indicates the energy transfer across different scales. In contrast, the baseline F-FNO shows higher rollout error and a mismatched structure function.

## 4 Experiments

The JAX [Bradbury et al., 2018] implementation of EddyFormer is available at `https://github.com/ASK-Berkeley/EddyFormer`, and a PyTorch [Paszke et al., 2019] version is provided via NVIDIA PhysicsNeMo (`https://github.com/NVIDIA/physicsnemo`). In §4.1 and §4.2, we consider two- and three-dimensional turbulent flows that are maintained by an artificial forcing. We evaluate the models using a set of comprehensive metrics (see §B for details). In §4.3, we benchmark EddyFormer on The Well [Ohana et al., 2025], a large-scale fluid simulation dataset under diverse flow conditions.

### 4.1 Homogeneous isotropic turbulence

We consider three-dimensional homogeneous isotropic turbulence, `Re94` . We use an isotropic forcing scheme on the lowest Fourier modes, which has been extensively used in turbulence modeling [Langford and Moser, 1999, Bazilevs et al., 2007, Yeung et al., 2015]. Specifically, a constant power forcing $\mathbf{f}(\mathbf{x})$ is applied on the lowest velocity modes $|\mathbf{k}| \leq 1$, in order to maintain the turbulence:

$$\frac{\partial \mathbf{u}}{\partial t} + \mathbf{u} \cdot \nabla \mathbf{u} = \nu \nabla^2 \mathbf{u} + \mathbf{f}(\mathbf{x}), \quad \mathbf{f}(\mathbf{x}) = \sum_{|\mathbf{k}| \leq 1, \mathbf{k} \neq 0} \frac{P_{\text{in}}}{E_1} \hat{\mathbf{u}}_{\mathbf{k}} e^{i\mathbf{k} \cdot \mathbf{x}}, \tag{13}$$

where $\nu = 0.01$ is the kinematic viscosity, $P_{\text{in}} = 1.0$ is the input power, and $E_1 = \frac{1}{2} \sum_{|\mathbf{k}| \leq 1} \hat{\mathbf{u}}_{\mathbf{k}} \cdot \hat{\mathbf{u}}_{\mathbf{k}}$ is the kinetic energy in lowest forced modes where $|\mathbf{k}| \leq 1$. The task is to predict the velocity field $\mathbf{u}$ after $\Delta t = 0.5$. This flow (see Fig. 1 for vorticity rendering) at equilibrium has Reynolds number $Re \approx 94$ at Taylor microscale. The dataset is recorded at $96^4$ resolution. We generate the dataset using a high-accuracy pseudo-spectral direct numerical simulation (DNS) at $384^3$ resolution, which corresponds to a Kolmogorov scale of $0.5\eta$. Detailed dataset and flow statistics can be found in §B.

For EddyFormer, we use $8^3$ elements and $13^3$ modes per element, resulting in approximately $\sim 0.1\%$ interpolation error on `Re94` . Detailed model architecture is described in Tab. 7. The LES filter size is set to $k_{\text{max}} = 5$ in order to balance accuracy and efficiency (see §C.2.1 for ablations on $k_{\text{max}}$).

**Direct prediction ("pure ML") benchmark.** We first benchmark EddyFormer in a purely learned setting, where the task is to directly predict the one-step velocity field $\mathbf{u}$ without using any coarse initial guess from a numerical solver. Baseline models include the classic ResNet [He et al., 2016], TF-Net [Wang et al., 2020] which performs filter-based splitting, the popular neural operators (FNO [Li et al., 2020] and F-FNO Tran et al. [2021]), and competitive Transformers for partial differential equations (GNOT [Hao et al., 2023] and AViT [McCabe et al., 2023]).

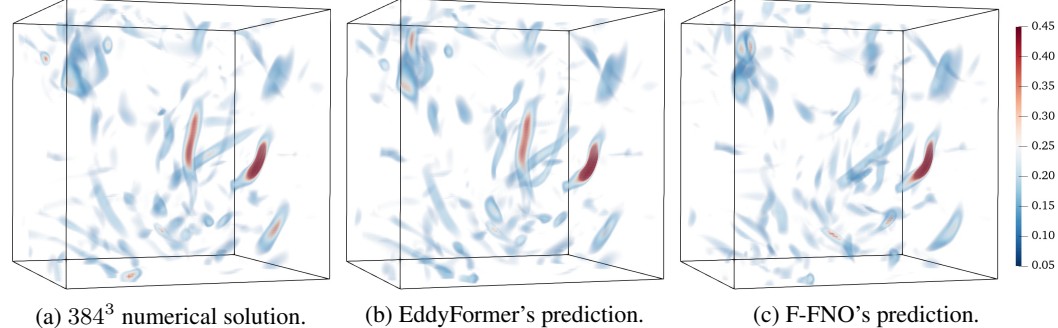

(a) $384^3$ numerical solution.     (b) EddyFormer's prediction.     (c) F-FNO's prediction.

Figure 4: Visualization of the Q-criterion for a `Re94` (§4.1) test sample at $t = 5$. The Q-criterion, Eqn. (19), identifies vortex structures where rotational motion dominates over strain, with the red volume in **(a)** reference DNS solution highlighting the vortex cores; **(b)** EddyFormer successfully captures both vortex cores, while **(c)** F-FNO fails to resolve the vortex at the center of the domain.

Every model is trained for 15k steps using the time-averaged one-step relative error,

$$L_\theta = \mathbb{E}_{t,\mathbf{u}_0}[\|\mathbf{u}_{t+\Delta t} - f_\theta(\mathbf{u}_t)\|/\|\mathbf{u}_{t+\Delta t}\|], \tag{14}$$

where $f_\theta$ is a neural network parameterized by $\theta$. Training details and loss curves for different models can be found in §C.2. Tab. 1 compares the one-step error and the training speed of each model. EddyFormer achieves lower test errors, combined with a compact model size and competitive training speed, outperforming all other baseline models.

Table 1: Direct prediction on `Re94` : model size, training time per iteration, and one-step relative error. EF (cheb) and EF (leg) stands for our model EddyFormer with Chebyshev and Legendre basis.

|  | ResNet | TF-Net | FNO | F-FNO | GNOT | AViT | EF (cheb) | EF (leg) |
|---|---|---|---|---|---|---|---|---|
| # of params. | 2.3M | 6.3M | 17.6M | 1.7M | 6.2M | 6.0M | 2.3M | 2.3M |
| Train speed | 1.2s/it | 3.0s/it | 1.5s/it | 1.5s/it | 7.4s/it | 1.0s/it | 2.7s/it | 1.8s/it |
| Test error | 87.2% | 27.7% | 23.1% | 22.5% | 69.7% | 26.6% | 8.61% | **8.52%** |

**Learned correction from a coarse numerical solver.** Next, we use the learned correction ("hybrid model") approach [Kochkov et al., 2021], which uses the output of a low-resolution DNS as an efficient initial solution and trains the model to predict the missing fine-scale information. This provides a good starting point for ML. The model additionally receives the output, $\mathbf{u}^*_{t+\Delta t}$, of a coarse solver, and predicts its error with respect to the high-resolution reference data $\mathbf{u}_{t+\Delta t}$. Similarly, the loss function is also defined as the time-averaged relative L2 error,

$$L_\theta = \mathbb{E}_{t,\mathbf{u}_0}\|\mathbf{u}_{t+\Delta t} - \mathrm{LC}_\theta(\mathbf{u}_t, \mathbf{u}^*_{t+\Delta t})\|/\|\mathbf{u}_{t+\Delta t}\|, \tag{15}$$

where $\mathrm{LC}_\theta(\mathbf{u}, \mathbf{u}^*) = \mathbf{u}^* + \alpha f_\theta(\mathbf{u}, \mathbf{u}^*)$ is the correction on the coarse solution $\mathbf{u}^*$. Here, we choose $\alpha = 10^{-5}$, which corresponds to an a priori estimate of the coarse solver's error. In practice, we find that model performance is not very sensitive to the $\alpha$ value.

For baseline comparison, we only consider F-FNO since it was the best baseline in the direct prediction benchmark. All models learn to correct DNS at $96^3$ resolution, which under-resolves the Kolmogorov scale $\eta$ by a ratio of 2. Each model is trained for 15k steps on four NVIDIA A100 GPUs, taking approximately one day wall-clock time to converge.

**Results.** Fig.3 shows the errors and invariant measures for both models, alongside DNS results at different resolutions. On the test set, EddyFormer matches the accuracy of DNS at $256^3$ resolution both the within trained window (Fig. 3a) and at later time steps (Fig. 3b), while accurately capturing the invariant structure functions over longer rollouts (Fig. 3c). EddyFormer also successfully predicts all characteristic vortices (Fig. 4), preserving both the shape and spatial coherence of highly rotational eddies.

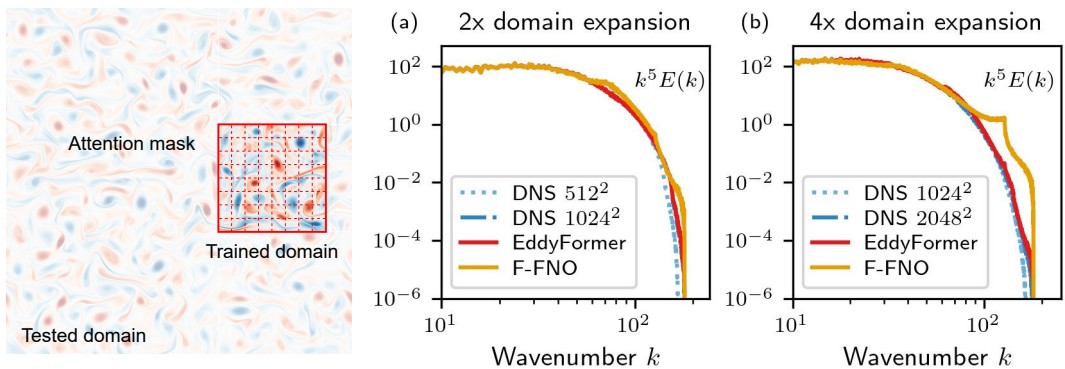

Figure 5: **(Left)** Illustration of how EddyFormer generalizes to larger domains. Trained on a smaller domain, EddyFormer predicts the flow field on a larger domain by applying an attention mask that ensures consistency with the training setup. The context window of each SEM token is constrained to a square of length $2\pi$, maintaining consistency between training and test times. **(Right)** Model evaluation on both 2x and 4x larger domains of KF4 (§4.2). The scaled energy spectra, Eqn. (17), of the EddyFormer's predictions align closely with the reference high-resolution DNS, while F-FNO does not capture this and shows deviation in high-frequency modes.

In addition to its accuracy, EddyFormer offers computational benefit. At test time, it only adds $\sim 50\%$ extra computational cost to the base DNS at $96^3$ resolution, accelerating the simulation by 30-fold on a single GPU (including the cost of the coarse DNS, as shown in Fig. 1), and is comparable to a high-accuracy DNS at $256^3$ resolution. This speed-up allows

Table 2: Simulation times of `Re94` and errors w.r.t the reference DNS at $384^3$ resolution. EddyFormer corrects the pseudo-spectral DNS at $96^3$ resolution.

|                 | DNS $96^3$ | DNS $256^3$ | EF (leg) |
|-----------------|------------|-------------|----------|
| Error at $t = 5$ | 55.9%      | 16.3%       | 18.2%    |
| Inference time  | 3.51s      | 152s        | 4.86s    |

for almost real-time high-accuracy simulations on `Re94`, enabling long-time ensemble rollouts that would otherwise be computationally expensive with high-resolution DNS.

## 4.2 Kolmogorov flow

We also consider the "Kolmogorov" flow `KF4`, another very popular model for studying turbulence in two dimensions. We use the scalar vorticity form of the Navier-Stokes equation:

$$\frac{\partial \omega}{\partial t} + \mathbf{u} \cdot \nabla \omega = \nu \nabla^2 \omega + k\cos(kx) - 0.1\omega, \tag{16}$$

where $\omega$ is the vorticity on $z$-axis. The energy is injected by a sinusoidal forcing $\mathbf{f}(x, y) = \sin(kx)\hat{\mathbf{e}}_y$ on periodic domain of length $2\pi$. Following Kochkov et al. [2021], we set the forcing mode $k = 4$, the viscosity $\nu = 10^{-3}$, and add a damping term $-0.1\mathbf{u}$ to prevent energy accumulation at the lowest modes [Chertkov et al., 2007]. The task is to predict the vorticity field $\omega$ after $\Delta t = 1$.

We set the forcing mode $k = 4$ and use the vorticity form of the equation. The resulting flow at equilibrium has a Reynolds number $Re \approx 800$ at Taylor microscale. The dataset is generated using a $4096^2$ pseudo-spectral DNS, which is more accurate than Kochkov et al. [2021] and Dresdner et al. [2022]. See §B for dataset details and simulation statistics.

All models are trained to correct DNS at a $256^2$ resolution. To control the propagation of errors over time, we incorporate a 5-step roll-out training approach, which minimizes accumulated error by differentiating through the DNS steps [Kochkov et al., 2021]. The roll-out loss function is applied during the fine-tuning stage, after the one-step loss has converged. See §C.1 for training details.

**Results.** In Fig. 8a and 8b, EddyFormer matches the accuracy of DNS at resolutions between $512^2$ and $1024^2$, both within the training window and beyond, in terms of relative error and correlations with the reference DNS. Over long rollouts (Fig. 8c), EddyFormer also accurately captures the invariant structure function whereas the baseline F-FNO shows significant deviation. Tab. 10 reports the kinetic energy, Eqn. (23), showing that EddyFormer conserves energy over extended time horizons.

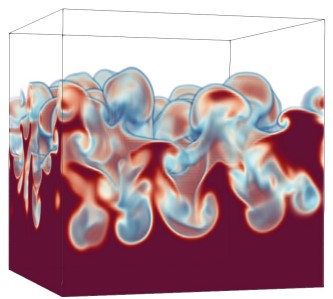 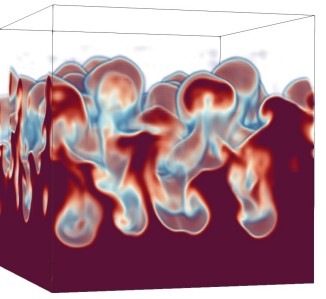 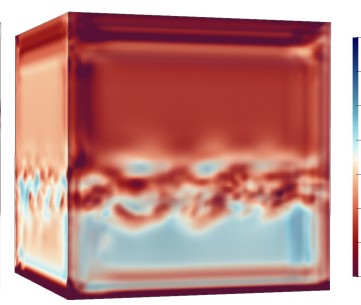

(a) Reference solution at $t = 30$.       (b) EddyFormer's prediction.       (c) FNO's prediction.

Figure 6: Density of a Rayleigh-Taylor instability test case. EddyFormer successfully predicts the structure of plumes due to buoyancy rise up to a long rollout of $t = 30$, while the baseline FNO fails to converge to a physical solution. See Fig. 16 for the full density evolution over time.

**Domain generalization.**   We evaluate EddyFormer's ability to generalize beyond the training domain by testing it on domains that are $4$ times larger. EddyFormer requires minimal modification (Fig. 5): we tile more elements proportionally to the domain expansion rate and apply a fixed $16 \times 16$ attention window to the SEMAttn block. This ensures that each spectral element remains the same size between training and test time, while still exchanging information within a fixed $[0, 2\pi]^2$ neighborhood. This also ensures that the computation cost only scales linearly with domain size.

Fig. 5 shows the scaled energy spectra on both 2x and 4x larger domains. EddyFormer consistently matches the energy spectrum of high-resolution DNS across all length scales, generalizing to larger scales without degradation in accuracy. In contrast, F-FNO, while performing well on the original domain, shows noticeable deviations in the high-frequency modes on the larger domains, indicating a loss in consistency as the domain size increases.

### 4.3  Turbulence beyond homogeneous isotropy

The Well [Ohana et al., 2025] is a large-scale physics simulation dataset that contains fluid flows under a diverse set of conditions. We benchmark EddyFormer on turbulence-related subsets, which include magnetohydrodynamics, shear layer, thermal convection, and buoyancy-driven turbulence. We follow their training protocols and describe experiment details in §C.3.

**Results.**   The VRMSE scores (Eqn. (25)) are reported in Tab. 3. Across all flow cases, EddyFormer achieves lower errors and consistently outperforms the baselines across a wide range of physically-driven turbulent flows. Fig. 6 visualizes the model predictions on the Rayleigh-Taylor instability problem. EddyFormer is able to accurately capture the development of turbulent plumes, while all baseline methods fail to converge.

Table 3: Performance on The Well dataset: time-averaged one-step VRMSE↓ scores on test sets. EF (cheb) and EF (leg) denote our EddyFormer model with Chebyshev and Legendre basis, respectively.

| Dataset name | FNO | T-FNO | U-net | CNextU-net | EF (cheb) | EF (leg) |
|---|---|---|---|---|---|---|
| MHD_64 | 0.3605 | 0.3561 | 0.1798 | 0.1633 | **0.1160** | 0.1240 |
| shear_flow | 1.189 | 1.472 | 3.447 | 0.8080 | 0.1123 | **0.0837** |
| rayleigh_benard | 0.8395 | 0.6566 | 1.4860 | 0.6699 | 0.2527 | **0.1854** |
| rayleigh_taylor_instab. | $> 10$ | $> 10$ | $> 10$ | $> 10$ | **0.1115** | 0.1243 |

**Conclusion.**   We introduced EddyFormer, an architecture inspired by the principles of large-eddy simulation, and designed for accurate and efficient turbulence simulation. EddyFormer demonstrates high accuracy with a significant acceleration on three-dimensional isotropic turbulence, as well as on flows driven by a wide range of conditions, ensuring stable rollouts over extended time periods. Future work will focus on extending EddyFormer to more complex flow domains and further scaling it for extreme-scale turbulence.

**Acknowledgements.** This work was supported by Laboratory Directed Research and Development (LDRD) funding under Contract Number DE-AC02-05CH11231. This research used resources of the National Energy Research Scientific Computing Center (NERSC), a Department of Energy User Facility. We thank Ningxiao Tao and Eric Qu for helpful discussions and feedback on earlier drafts.

**Broader impacts.** The development of EddyFormer has the potential to impact both scientific research and practical applications. By accelerating turbulence simulations, it can aid in solving complex fluid dynamics problems across fields such as climate modeling, aerospace engineering, and energy systems, where understanding turbulence is critical. Additionally, the use of machine learning to improve simulation efficiency opens new possibilities for real-time simulations, making it feasible to explore more dynamic and complex systems. However, as with any AI-driven technology, it is important to consider potential risks, such as over-reliance on model accuracy without sufficient validation or unintended biases in training data. Moreover, as simulation tools become more accessible, there is a responsibility to ensure their ethical and equitable use, particularly in areas like environmental modeling and resource management, where decisions based on inaccurate simulations could have widespread societal impacts.

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

# A  Discussions on related works

We provide a broader overview of related work in ML for fluid dynamics, covering physics-informed learning, spectral methods, Transformer-based models, and reduced-order modeling.

**ML for fluids.**    Except for the data-driven approaches discussed in §1, another line of work focuses on developing large pretrained models that can generalize across a wide range of flow conditions, geometries, and Reynolds numbers [McCabe et al., 2023, Herde et al., 2024, Rahman et al., 2024]. These models are trained on diverse simulation datasets and aim to serve as universal surrogates for fluid dynamics. In parallel, generative approaches have been proposed to capture the stochasticity and variability of turbulent flows. Lienen et al. [2023] apply diffusion models to synthesize physically plausible trajectories in a zero-shot setting, while Boral et al. [2023] introduce neural stochastic differential equations to model ensembles of turbulent trajectories.

Another popular direction involves physics-informed machine learning [Karniadakis et al., 2021], which integrates prior knowledge of problem during training. Physics-Informed Neural Networks (PINNs) [Raissi et al., 2019] incorporates the residuals of the governing equations into the loss function. This framework allows for training with sparse or no supervision. Works in this direction seek to address challenges such as training strategy [Krishnapriyan et al., 2021, Wang et al., 2022], residual estimation [McClenny and Braga-Neto, 2023, Du et al., 2024], and constraint enforcement [Négiar et al., 2022, Chalapathi et al., 2024]. Recent advances also demonstrated that PINNs are able to simulate large-scale three-dimensional turbulence [Wang et al., 2025]. However, the training cost of PINNs, and solving for one problem at a time, makes their efficiency worse compared to state-of-the-art numerical simulations.

**Spectral-based models.**    Spectral methods [Canuto, 2007] provide powerful inductive biases for neural fluid simulation. Li et al. [2020] introduced Fourier neural operator (FNO), which is based on spectral convolutions in the Fourier domain. Neural operators have demonstrated accuracy and efficiency advantages over traditional convolutional networks. Subsequent works extended neural operators by either improving the spectral kernel's design [Tran et al., 2021, Fanaskov and Oseledets, 2023, Helwig et al., 2023], or by learning completely in the spectral domain [Du et al., 2024, Xia et al., 2023]. These models have shown promising results on low-dimensional problems; however, due to the inefficiency of parameterizing and learning large Fourier convolution kernels [Qin et al., 2024], their application to large-scale three-dimensional turbulence has been limited.

Moreover, they face challenges on irregular domain geometries since spectral methods are inherently homogeneous. To overcome this, several extensions have been proposed that incorporate geometry more explicitly [Li et al., 2023b,c]. However, these methods still encounter limitations in accurately capturing complex geometries, as well as in applying to problems with a wide range of length scales.

**Transformers for spatiotemporal modeling.**    There has been an increased interest in exploring the Transformer architecture for fluid simulation, and more generally, for solving partial differential equations. One line of work [Dang et al., 2022, McCabe et al., 2023, Herde et al., 2024] applied well-established Vision Transformer (ViT) architectures to this task. However, due to the quadratic complexity of the self-attention mechanism, their scalability is limited by the spatial resolution of the solution field, especially for the resolution of three-dimensional turbulent flows.

Several approaches have been proposed to address this challenge: **(a) simplified attention**: Cao [2021] draw connections between the attention mechanism and kernel convolution, providing insights into reducing computational complexity. GNOT [Hao et al., 2023], FactFormer [Li et al., 2023a], OFormer [Li et al., 2022], and AViT [McCabe et al., 2023] applied different simplifications to the attention mechanism, enabling linear or almost-linear computational complexity; **(b) grouped tokens**: this approach reduces the number of tokens involved in the attention mechanism by grouping relevant mesh points together. UPT [Alkin et al., 2024] uses sub-sampling supernodes in combination with a perceiver-style architecture, while Transolver [Wu et al., 2024, Luo et al., 2025] operates on slice-based physical tokens. While these approaches have shown promise for moderate-scale problems, their effectiveness and accuracy on large-scale turbulence problems is unclear.

**Reduced-order modeling.**    Reduced Order Modeling (ROM) provides a framework for approximating high-fidelity simulations with lower computational cost. Classical methods such as Proper Orthogonal Decomposition (POD) [Berkooz et al., 1993] have been extensively used as low-dimensional representations. Data-driven approaches, including Dynamic Mode Decomposition [Williams et al.,

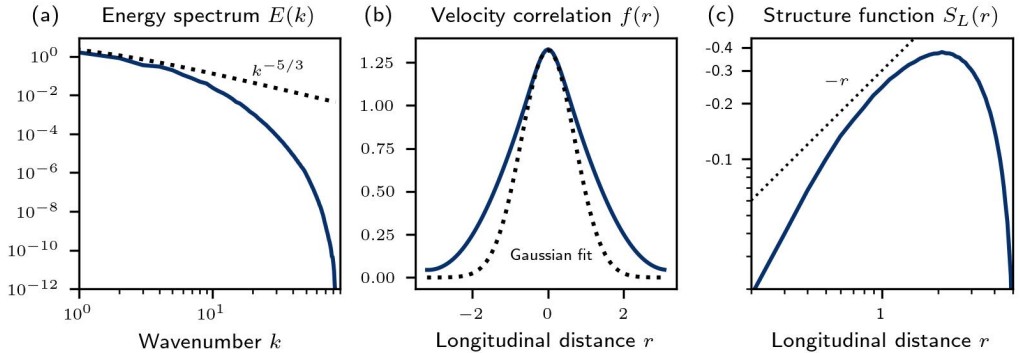

Figure 7: Energy spectrum, velocity correlation, and structure function [Pope, 2001] of `Re94`. The energy spectrum follows the $-5/3$ scaling law, indicating inertial-range turbulence. The non-Gaussian velocity correlation shows intermittent behavior. Additionally, the structure function exhibits scaling consistent with Kolmogorov's theory, further confirming the presence of turbulent dynamics.

2015, Kutz et al., 2016] and Operator Inference [Peherstorfer and Willcox, 2016], have expanded the scope of reduced-order modeling by constructing surrogate models directly from data. Recently, autoencoders has been employed to construct nonlinear reduced-order models [Lee and Carlberg, 2020, Pant et al., 2021, Kim et al., 2022]. These approaches aim to learn low-dimensional manifolds that capture the essential dynamics of complex systems beyond linear methods. However, these methods often require complete information of the solution regime; consequently, their accuracy and generalization usually remain problem-dependent.

## B  Dataset details

In this section, we provide details on the two and three-dimensional turbulence dataset used in our experiments. Tab. 4 and Tab. 5 contains simulation parameters and flow statistics in the dataset. Fig. 7 plots the energy spectrum of each flow, averaged across the whole test set.

**Initial condition.**   The lowest Fourier modes $|\mathbf{k}| < 16$ of every initial velocity field are randomly initialized according to the $k^{-5/3}$ spectrum. The initial condition is then burned in for a fixed time $t_0$, which is chosen empirically so that energy injection and dissipation reach equilibrium.

**Solver details.**   We use pseudo-spectral method to generate the high-accuracy dataset as well as the low-resolution input for learned correction on top of it. Smooth dealiasing [Hou and Li, 2007] is used during advection in order to keep more modes than the common $2/3$ rule. We use fourth-order Runge-Kutta scheme for time-stepping, coupled with a second-order implicit diffusion step. The time step is determined dynamically by fixing Courant–Friedrichs–Lewy (CFL) number below $C_{\max} = 1$.

**Dataset.**   The dataset is downsampled onto uniform grids using FFT at intervals of $\Delta t$, over a total duration $t$. The test set contains 16 longer trajectories: $t = 50s$ for `KF4` and $t = 20s$ for `Re94`.

Table 4: Simulation parameters of KF4 (§4.2) and Re94 (§4.1).

| Name | KF4 | Re94 |
|---|---|---|
| Domain $\Omega$ | $[0, 2\pi]^2$ | $[0, 2\pi]^3$ |
| Burn-in time $t_0$ | $40s$ | $20s$ |
| Record time $\Delta t$ | $0.125s$ | $0.1s$ |
| Trajectory length $t$ | $10s$ | $5.0s$ |
| Solver resolution | $4096^3$ | $384^3$ |
| Dataset resolution | $256^2$ | $96^3$ |
| Dataset size | 89GB | 1TB |

Table 5: Turbulence statistics of KF4 (§4.2) and Re94 (§4.1).

| Name | KF4 | Re94 |
|---|---|---|
| Viscosity $\nu$ | 0.001 | 0.01 |
| Kinetic energy $E_k$ | 2.0061 | 3.6023 |
| Dissipation rate $\epsilon$ | 0.0419 | 0.9749 |
| Velocity r.m.s. $\langle v \rangle_2$ | 1.1539 | 1.5493 |
| Taylor microscale $\lambda$ | 0.6939 | 0.6090 |
| Kolmogorov scale $\eta$ | 0.0124 | 0.0318 |
| Reynolds no. $Re_\lambda$ | 799.76 | 94.40 |

**Evaluation metrics.**   We report standard error metrics, such as relative $L_2$ error, as well as other invariant flow statistics when evaluating all models and numerical solvers. Specifically, let $\mathbf{u}$ be the reference solution and $\mathbf{u}'$ be the prediction, the following metrics are reported:

1. Relative L2 error $\|\mathbf{u}-\mathbf{u}'\|_2/\|\mathbf{u}\|_2$. Relative error is used when $\mathbf{u}$ and $\mathbf{u}'$ are highly correlated, i.e., during early steps in the rollout. It is also used as the training objective (see Eqn. (15)).

2. Point-wise correlation $\langle\mathbf{u},\mathbf{u}'\rangle/\|\mathbf{u}\|_2^2$. Correlation is used in later solution rollout, when $\mathbf{u}$ and $\mathbf{u}'$ are statistically correlated, but relative error is too large to be a meaningful metric.

3. Energy spectrum $E(k)$,

$$E(k) = \int_{|\mathbf{k}|=k} \frac{1}{2}|\hat{\mathbf{u}}(\mathbf{k})|^2 d\mathbf{k}. \tag{17}$$

The energy spectrum describes how turbulent energy is distributed across different length scales. In the inertial range, $E(k)$ is expected to follow Kolmogorov's 5/3 law, i.e., $E(k) \propto k^{-5/3}$. Accurately reproducing the energy spectrum is necessary for a model to yield physically meaningful predictions; even after the model's prediction becomes decorrelated from the reference, $E(k)$ remains critical because it still reflects the turbulence cascade.

4. Two-point third-order longitudinal structure function $S_L$,

$$S_L(r) = \langle([\mathbf{u}(\mathbf{x}+\mathbf{r}) - \mathbf{u}(\mathbf{x})] \cdot \frac{\mathbf{r}}{r})^3\rangle_{|\mathbf{r}|=r}. \tag{18}$$

The structure function $S_L(r)$ measures the statistical asymmetry of the velocity differences across a distance $r$, indicating the downscale energy transfer across turbulence scales. It is hypothesized to follow the Kolmogorov's 4/5 law, i.e., $S_L(r) = -\frac{4}{5}\epsilon r$, where $\epsilon$ is the dissipation rate. However, in two-dimensional turbulence where inverse energy transfer, i.e., "backscattering", occurs, $S_L$ exhibits more complex behavior [Kraichnan, 1967, Kraichnan and Montgomery, 1980, Xie and Bühler, 2018]. Evaluating the two-point structure function on a model's prediction tests whether the it preserves the correct inter-scale statistics.

5. The Q-criterion $q$,

$$q = \frac{1}{2}(\|\Omega\|^2 - \|S\|^2), \tag{19}$$

where $S = \frac{1}{2}(\nabla u + \nabla u^T)$ and $\Omega = \frac{1}{2}(\nabla u - \nabla u^T)$ are the strain rate and rotation tensor, respectively. The Q-criterion is a widely used diagnostic metric [Jeong and Hussain, 1995] for identifying vortical structures in turbulent flows. Regions where $q > 0$ indicate local dominance of rotational motion over strains, which typically corresponds to coherent vortex cores. It provides a frame-invariant and physically interpretable way to visualize and quantify vortical structures in complex turbulent flows. We use the Q-criterion to visually compare the vortex cores in the reference DNS solution and EddyFormer's prediction.

## C   Experiment details

**Hyperparameters.**   For 2D and 3D problems, EddyFormer uses the hyperparameters described in Tab. 6 and Tab. 7, respectively. We consider both Chebyshev and Legendre basis as the orthogonal polynomials in SEM basis for most problems, which is indicated by EddyFormer (cheb) and Eddy-Former (leg), respectively. We use the Adam optimizer with a learning rate of $10^{-3}$. For homogeneous turbulence §4.1 and §4.2, we use a batch size of 16, while a batch size of 64 is used in §4.3.

Table 6: EddyFormer architecture (2D).

| Stream | LES | SGS |
|---|---|---|
| Number of layers $L$ | 10 | |
| Hidden dimension $d$ | 32 | |
| Mesh size $H$ | $16^2$ | |
| Mode $k_{\max}$ and $M$ | 4 | 24 |
| Kernel mode $m$ | 4 | 24 |
| Kernel size $s$ | $\pi/4$ | $\pi/4$ |
| Number of heads $n_{\text{head}}$ | 8 | — |
| Head dimension $d_{\text{head}}$ | 16 | — |

Table 7: EddyFormer architecture (3D).

| Stream | LES | SGS |
|---|---|---|
| Number of layers $L$ | 4 | |
| Hidden dimension $d$ | 32 | |
| Mesh size $H$ | $8^3$ | |
| Mode $k_{\max}$ and $M$ | 5 | 13 |
| Kernel mode $m$ | 5 | 13 |
| Kernel size $s$ | $\pi/4$ | $\pi/4$ |
| Number of heads $n_{\text{head}}$ | 4 | — |
| Head dimension $d_{\text{head}}$ | 32 | — |

**Implementation details.** EddyFormer is implemented using the JAX framework [Bradbury et al., 2018]. We implement our own three-dimensional pseudo-spectral solver for all data generations and learned correction used in our experiments. We acknowledge the JAX-CFD [Dresdner et al., 2022] and NSM [Du et al., 2024] codebases, which we referenced and built upon in our implementation.

**Implementation of SEMConv.** The SEM-based convolution, Eqn. (8), evaluates the convolution between a Fourier kernel and features on collocation points. For simplicity, here we only consider the implementation of 1D convolution. Its extension to multi-dimensional case is straightforward due to the use of factorized kernels. Specifically, the SEMConv performed along $x$ axis is defined by,

$$\text{SEMConv}(\mathbf{u})(x') = \int_{|x| \leq s/2} \mathbf{k}(x)\mathbf{u}(x' - x)^T dx. \tag{20}$$

We use quadrature rules to directly compute the convolution on each collocation point. Specifically, Clenshaw–Curtis and Gauss-Lobatto-Legendre quadratures are use for Chebyshev and Legendre-based SEM bases, respectively. For each collocation point $x_h^{(n)}$ in the $h$'th element, the integral is the sum of the quadratures in neighbor elements,

$$\text{SEMConv}(\mathbf{u})(x_h^{(n)}) \approx \sum_{|h'-h| \leq S} \sum_{m \leq M} w_{h'}^{(m)} \mathbf{k}(x_{h'}^{(m)} - x_h^{(n)}) \mathbf{u}(x_{h'}^{(m)})^T, \tag{21}$$

where $S = \lceil \frac{s}{2\Delta} \rceil$ is the half-width of the neighbors within the convolution window $s$, and $\{w_{h'}^{(m)}\}_m$ are the quadrature weights on element $\Omega_{h'}$, which can be pre-computed using routines from standard packages like QUADPACK [Virtanen et al., 2020]. Below is a pseudo-code of our implementation.

---
**Algorithm 1:** SEMConv Implementation (1D)
---
**Input:** Feature field $\mathbf{u}$; Fourier convolution kernel $\mathbf{k}(x)$
**Output:** Convolved feature field SEMConv($\mathbf{u}$)
$S \leftarrow \lceil s/2\Delta \rceil$ ;                                   // Number of neighbors
$w \leftarrow \text{QuadratureWeights}()$ ;                      // Precomputed weights
**foreach** *collocation point* $x_h^{(n)}$ **do**
   $y \leftarrow 0$ ;                                                 // Output at $x_h^{(n)}$
   **for** $h' = h - S$ **to** $h + S$ **do**
      **for** $m = 0$ **to** $M$ **do**
         $w \leftarrow w_{h'}^{(m)}$ ;                          // Quadrature weight
         $k \leftarrow \mathbf{k}(x_{h'}^{(m)} - x_h^{(n)})$ ;        // Evaluate convolution kernel
         $u \leftarrow \mathbf{u}(x_{h'}^{(m)})$ ;               // Evaluate input feature
         $y \mathrel{+}= w \cdot k \cdot u^T$ ;
   $\text{SEMConv}(\mathbf{u})(x_h^{(n)}) \leftarrow y$

---

**Computation cost of SEMConv.** The computational cost of Algorithm 1 consists of two main components: evaluating the kernel $\mathbf{k}$ on collocation points and performing quadrature over the input field. Under the assumption of a uniform mesh, the kernel only needs to be evaluated at $MS$ distinct offsets, which is equivalent to performing an $MS \times MS$-size matrix multiplication. The quadrature step involves a weighted dot product across neighbors, which costs $\mathcal{O}(M^2 S)$ per element. Therefore, the total cost scales with the number of resolved modes, with a sublinear factor of $\mathcal{O}(MS)$.

## C.1 Two-dimensional turbulence KF4

**Learned correction.** All models are trained to correct a DNS at $256^2$ resolution. Each model is first trained using the one-step loss, Eqn. (15), for 30k steps with a learning rate of $10^{-3}$. In order to control the roll-out error over time, we also differentiate through the DNS steps and minimize accumulated error as Kochkov et al. [2021] did. Specifically, $\text{LC}_\theta^{(n)}$ represents the $n$-step prediction roll-out using learned corrections, and the $N$-step loss function is defined as follows:

$$\mathbb{E}_{t,\mathbf{u}_0}[\frac{1}{N} \sum_{n \leq N} \|\mathbf{u}_{t+n\Delta t} - \text{LC}_\theta^{(n)}(\mathbf{u}_t)\| / \|\mathbf{u}_{t+n\Delta t}\|], \tag{22}$$

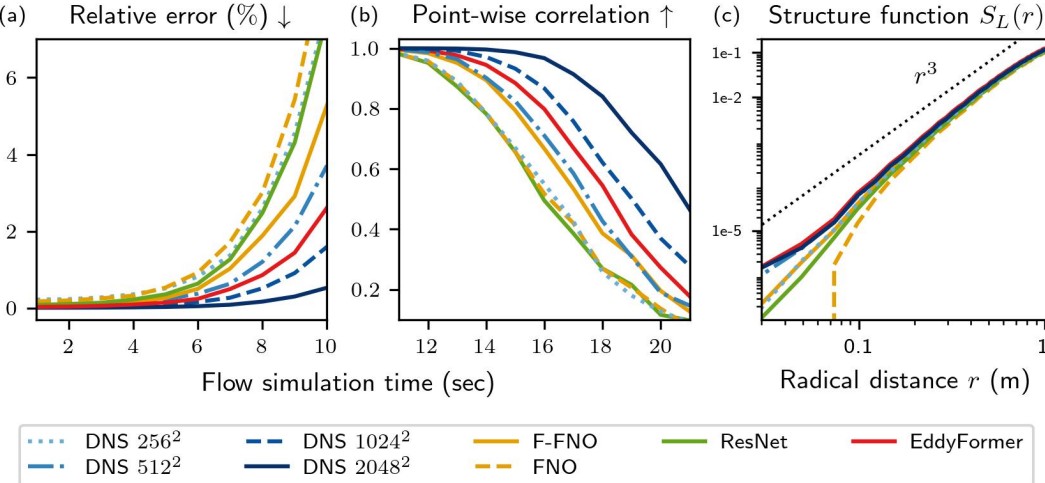

Figure 8: Performance of models and DNS at different resolutions on KF4 (§4.2), a two-dimensional fluid flow example ("Kolmogorov" flow). The models learn to correct DNS at $256^2$ resolution, and the metrics here are relative to a reference DNS at $4096^2$ resolution. EddyFormer achieves the accuracy of DNS between $512^2$ and $1024^2$ resolution in terms of **(a)** relative L2 error for $t \leq 10$ and **(b)** point-wise correlation for $10 \leq t \leq 20$; for **(c)** longer rollouts up to $50$ seconds, EddyFormer also accurately captures the third-order structure function, Eqn. (18), which characterizes the forward and inverse energy transfer in fine length-scales.

However, as reported by Dresdner et al. [2022], directly minimizing multi-step loss is difficult and unstable. Therefore, we choose $N = 5$ and only fine-tuned using the 5-step loss for another 5k steps with a reduced learning rate of $10^{-5}$, after the one-step training converged.

**Baseline models.** We consider ResNet [He et al., 2016], FNO [Li et al., 2020], and F-FNO [Tran et al., 2021] as baseline models for comparison. For ResNet, we adopt the standard ResNet-18 architecture. For both FNO and F-FNO, we use $64^2$ modes, with 10 layers and 32 hidden dimensions, to ensure a fair comparison with EddyFormer. All models, except for FNO, have a model size of the order of 10MB.

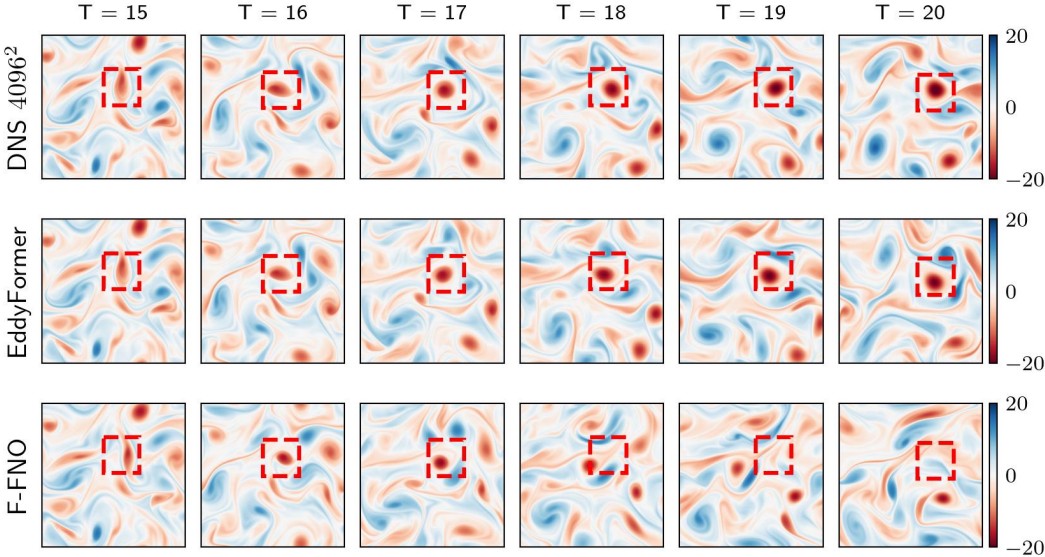

Figure 9: Vorticity visualization of a test sample on KF4 (§4.2). EddyFormer accurately tracks the dynamics of the vortex core (indicated by the red square) up to $t = 20$, while the baseline F-FNO fails to resolve it over the long rollout.

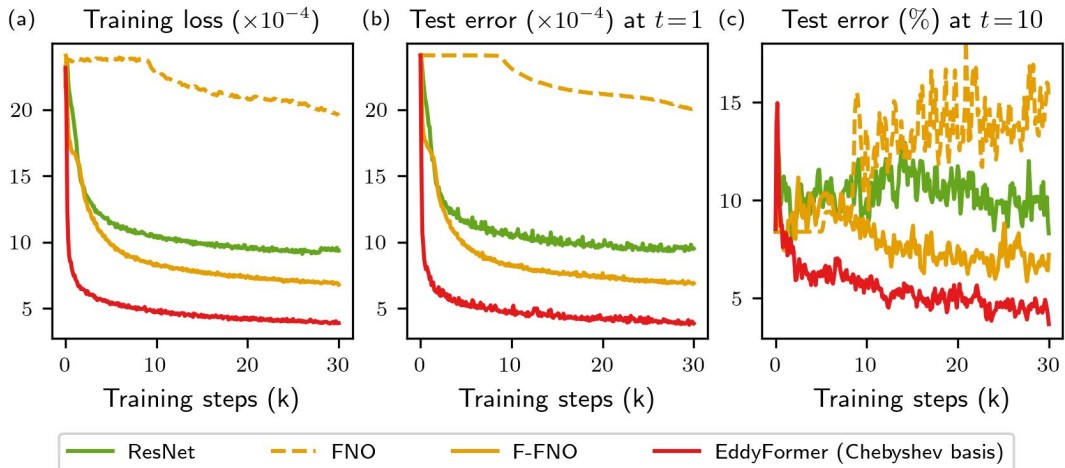

Figure 10: **(a)** Training loss, and **(b)** and **(c)** test error curves on KF4 (§4.2). EddyFormer converges significantly faster than all baselines, while achieving almost a $50\%$ reduction in converged loss.

For reference, we also include results reported by previous work [Kochkov et al., 2021], which were obtained by correcting a lower-accuracy finite-volume DNS at $64^2$ resolution. For a fair comparison, we reproduced their result by correcting our high-accuracy pseudo-spectral solver using a ResNet.

**Results.** Fig. 10 shows the training loss as well as test errors at different times. Tab. 8 and Tab. 9 report the relative L2 error and point-wise correlation metrics on test set. EddyFormer is able to converge faster than the baselines, consistently achieving lower test error. Fig. 9 visualizes the evolution of a vortex core, showing that EddyFormer captures its development up to $t = 20$.

Table 8: Relative L2 error (%) on KF4 test set with learned correction of DNS $256^2$.

|  | $t{=}1$ | $t{=}2$ | $t{=}3$ | $t{=}4$ | $t{=}5$ | $t{=}6$ | $t{=}7$ | $t{=}8$ | $t{=}9$ | $t{=}10$ |
|---|---|---|---|---|---|---|---|---|---|---|
| DNS $256^2$ | 0.23 | 0.24 | 0.28 | 0.36 | 0.52 | 0.82 | 1.38 | 2.59 | 4.55 | 7.79 |
| DNS $512^2$ | 0.09 | 0.10 | 0.12 | 0.16 | 0.23 | 0.38 | 0.64 | 1.21 | 2.13 | 3.71 |
| DNS $1024^2$ | 0.04 | 0.04 | 0.05 | 0.07 | 0.10 | 0.16 | 0.27 | 0.52 | 0.92 | 1.60 |
| Kochkov et al. | 0.63 | 1.38 | 2.53 | 4.54 | 8.09 | 14.7 | 24.3 | 36.5 | 56.4 | 74.5 |
| FNO | 0.17 | 0.20 | 0.28 | 0.40 | 0.71 | 1.19 | 2.06 | 3.58 | 6.59 | 11.4 |
| ResNet | 0.08 | 0.10 | 0.14 | 0.22 | 0.36 | 0.63 | 1.27 | 2.47 | 4.30 | 7.58 |
| F-FNO | 0.06 | 0.07 | 0.10 | 0.15 | 0.27 | 0.50 | 1.03 | 1.88 | 2.91 | 5.28 |
| EddyFormer | 0.03 | 0.04 | 0.06 | 0.09 | 0.14 | 0.24 | 0.49 | 0.86 | 1.44 | 2.61 |

Table 9: Point-wise correlation on KF4 test set with learned correction of DNS $256^2$.

|  | $t{=}11$ | $t{=}12$ | $t{=}13$ | $t{=}14$ | $t{=}15$ | $t{=}16$ | $t{=}17$ | $t{=}18$ | $t{=}19$ | $t{=}20$ |
|---|---|---|---|---|---|---|---|---|---|---|
| DNS $256^2$ | 0.98 | 0.96 | 0.89 | 0.78 | 0.67 | 0.55 | 0.43 | 0.26 | 0.18 | 0.13 |
| DNS $512^2$ | 1.00 | 0.99 | 0.96 | 0.90 | 0.83 | 0.71 | 0.59 | 0.43 | 0.31 | 0.19 |
| DNS $1024^2$ | 1.00 | 1.00 | 0.99 | 0.97 | 0.93 | 0.87 | 0.76 | 0.62 | 0.50 | 0.37 |
| Kochkov et al. | 0.69 | 0.51 | 0.36 | 0.24 | 0.15 | 0.09 | 0.09 | 0.02 | $-0.01$ | 0.02 |
| FNO | 0.98 | 0.96 | 0.89 | 0.79 | 0.66 | 0.52 | 0.42 | 0.27 | 0.21 | 0.14 |
| ResNet | 0.98 | 0.95 | 0.87 | 0.78 | 0.66 | 0.50 | 0.39 | 0.27 | 0.22 | 0.12 |
| F-FNO | 0.99 | 0.98 | 0.95 | 0.89 | 0.79 | 0.67 | 0.53 | 0.39 | 0.31 | 0.20 |
| EddyFormer | 1.00 | 0.99 | 0.98 | 0.94 | 0.88 | 0.80 | 0.67 | 0.55 | 0.38 | 0.27 |

**Discussions on domain generalization.** This expansion task (Fig. 5) is conceptually different from super-resolution. While super-resolution [Li et al., 2020] aims to recover finer-scale details on a fixed physical domain, domain expansion keeps the grid spacing and increases only the physical extent of the flow, testing whether the model has overfitted to domain-specific flow dynamics. Neural operators have no corresponding expansion mechanism. We tried two ad-hoc work-arounds: rescaling the appended spatial coordinates and stretching the learned convolution kernels. Neither produced comparable results: Fig. 5 shows that F-FNO, by rescaling the appended coordinates, fails to match the energy spectra on the expanded domains.

**Energy conservation.** In addition to evaluating the statistical properties of model's long-term prediction, we also assess the model's ability to conserve total kinetic energy, $E_{\text{tot}}$, over time:

$$E_{\text{tot}}(T) = k|_{t=0} + \int_0^T I(t) - \epsilon(t)dt, \tag{23}$$

where the kinetic energy $k = \langle \frac{1}{2}\mathbf{u}^2 \rangle$, the dissipation rate $\epsilon = 2\nu \langle \|S\|_2^2 \rangle$ with $S = \frac{1}{2}(\nabla u + \nabla u^T)$ denoting the strain-rate tensor, and the injection rate $I = \langle \mathbf{f} \cdot \mathbf{u} \rangle$ given a forcing term $\mathbf{f}$. This cumulative energy accounts for the balance of injection and dissipation and should remain constant in time. Even though energy conservation is a necessary condition for a physically correct model and is implicitly present in the training data, the models are not explicitly informed of this constraint and must learn to conserve energy purely from data.

Tab. 10 reports the time-averaged total kinetic energy over a test trajectory. The results indicate that EddyFormer preserves the energy balance over long rollouts, whereas the baseline F-FNO exhibits significant and systematic energy dissipation over time.

Table 10: Time-averaged total kinetic energy $E_{\text{tot}}(t)$ of a trajectory in KF4 test set.

| Time interval | $[0, 5]$ | $[5, 10]$ | $[10, 15]$ | $[15, 20]$ | $[20, 25]$ | $[25, 30]$ | $[30, 35]$ | $[35, 40]$ | $[40, 45]$ | $[45, 50]$ |
|---|---|---|---|---|---|---|---|---|---|---|
| EddyFormer | 1.728 | 1.668 | 1.692 | 1.660 | 1.712 | 1.736 | 1.695 | 1.694 | 1.706 | 1.670 |
| F-FNO | 1.728 | 1.668 | 1.688 | 1.588 | 1.650 | 1.623 | 1.524 | 1.518 | 1.502 | 1.511 |

### C.2 Three-dimensional turbulence `Re94`

The `Re94` case is adopted from Bazilevs et al. [2007]. The forcing term $\mathbf{f}(\mathbf{x})$,

$$\mathbf{f}(\mathbf{x}) = \sum_{|\mathbf{k}| \leq 1, \mathbf{k} \neq 0} \frac{P_{\text{in}}}{E_1} \hat{\mathbf{u}}_{\mathbf{k}} e^{i\mathbf{k} \cdot \mathbf{x}}, \tag{24}$$

injects energy uniformly across all directions. Here, $\hat{\mathbf{u}}_{\mathbf{k}}$ is the $\mathbf{k}$'th Fourier coefficient of the velocity.

**Direction prediction ("pure ML") benchmark.** For baseline comparisons, we adopt ResNet, FNO, and F-FNO in 3D without modifying their original architectures. For FNO, we reduce the number of modes to $16^3$ in order to avoid exceeding the model size constraint. In addition, we also consider TF-Net [Qin et al., 2024], GNOT [Hao et al., 2023], and AViT [McCabe et al., 2023] as additional baselines. We use their default architectures and adapt them to handle three-dimensional input fields. The number of trainable parameters for each model is reported in Tab. 1.

All models are trained for 15k steps using one-step loss. Fig. 11 shows the training loss of different baseline models. EddyFormer converges significantly faster and consistently achieving lower test error than state-of-the-art neural operators and Transformers. In these baselines, F-FNO is the best model, yet EddyFormer achieves almost $3\times$ lower error.

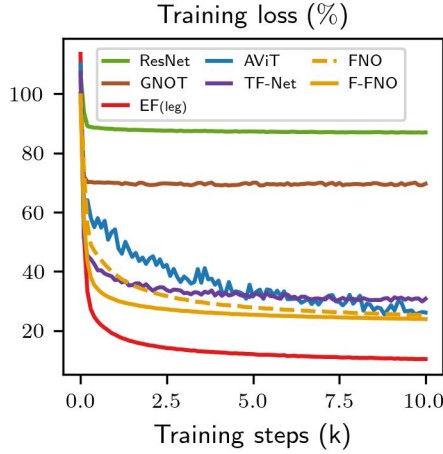

Figure 11: Direct prediction loss curves `Re94` (§4.1). EF (leg) denotes EddyFormer with Legendre basis.

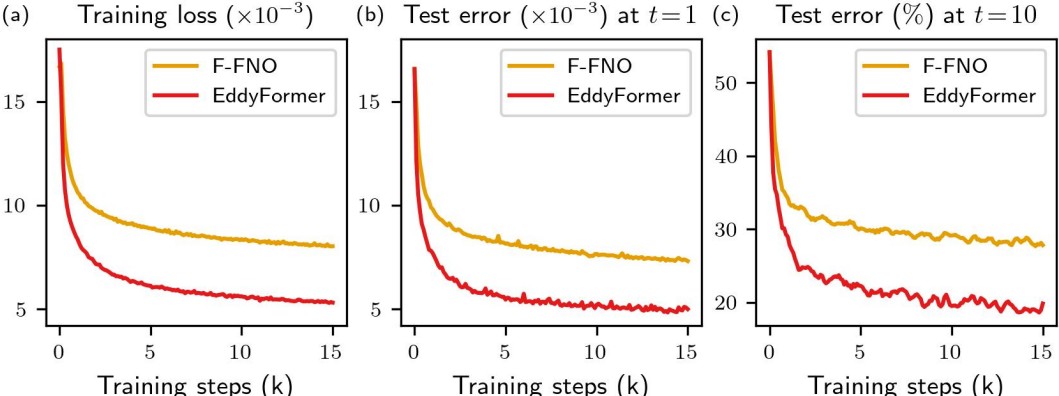

Figure 12: **(a)** Training loss, and **(b)** and **(c)** test error curves on `Re94` (§4.1) with learned correction of DNS at $96^3$ resolution. EddyFormer consistently achieves lower error on both training and test set.

**Learned correction.** Both F-FNO and EddyFormer are trained using one-step loss, Eqn. (15), for 15k steps with a learning rate of $10^{-3}$. Fig. 12 shows the training loss as well as test errors at different times. Tab. 11 and Tab. 12 report the relative L2 error and point-wise correlation metrics on test set.

Table 11: Relative L2 error (%) on `Re94` test set with learned correction of DNS $96^3$.

|  | $t=0.5$ | $t=1.0$ | $t=1.5$ | $t=2.0$ | $t=2.5$ | $t=3.0$ | $t=3.5$ | $t=4.0$ | $t=4.5$ | $t=5.0$ |
|---|---|---|---|---|---|---|---|---|---|---|
| DNS $96^3$ | 1.53 | 3.34 | 5.64 | 8.74 | 13.0 | 18.7 | 26.0 | 34.8 | 44.8 | 55.9 |
| DNS $128^3$ | 1.04 | 2.28 | 3.88 | 6.08 | 9.1 | 13.3 | 19.0 | 26.2 | 35.0 | 45.5 |
| DNS $192^3$ | 0.53 | 1.17 | 2.00 | 3.17 | 4.84 | 7.16 | 10.4 | 14.9 | 20.9 | 28.8 |
| DNS $256^3$ | 0.27 | 0.60 | 1.02 | 1.62 | 2.48 | 3.70 | 5.44 | 7.92 | 11.3 | 16.3 |
| F-FNO | 0.68 | 1.18 | 1.87 | 2.93 | 4.44 | 6.60 | 9.71 | 14.0 | 19.8 | 27.4 |
| EddyFormer | 0.46 | 0.78 | 1.23 | 1.89 | 2.80 | 4.20 | 6.23 | 9.27 | 13.1 | 18.2 |

Table 12: Point-wise correlation on `Re94` test set with learned correction of DNS $96^3$.

|  | $t=5.5$ | $t=6.0$ | $t=6.5$ | $t=7.0$ | $t=7.5$ | $t=8.0$ | $t=8.5$ | $t=9.0$ | $t=9.5$ | $t=10.$ |
|---|---|---|---|---|---|---|---|---|---|---|
| DNS $96^3$ | 0.78 | 0.71 | 0.63 | 0.55 | 0.47 | 0.39 | 0.30 | 0.23 | 0.16 | 0.09 |
| DNS $128^3$ | 0.84 | 0.78 | 0.70 | 0.62 | 0.54 | 0.45 | 0.36 | 0.28 | 0.20 | 0.13 |
| DNS $192^3$ | 0.93 | 0.88 | 0.82 | 0.75 | 0.67 | 0.60 | 0.51 | 0.43 | 0.34 | 0.25 |
| DNS $256^3$ | 0.97 | 0.95 | 0.91 | 0.86 | 0.80 | 0.74 | 0.67 | 0.59 | 0.51 | 0.43 |
| F-FNO | 0.93 | 0.89 | 0.83 | 0.76 | 0.69 | 0.62 | 0.54 | 0.46 | 0.38 | 0.30 |
| EddyFormer | 0.97 | 0.94 | 0.90 | 0.85 | 0.78 | 0.71 | 0.64 | 0.56 | 0.48 | 0.41 |

### C.2.1 Ablation studies

In order to better understand EddyFormer and the right model design choices, we perform various ablation studies using the direct prediction ("pure ML") benchmark on `Re94` . First, we consider the following modifications to isolate the model components:

- **EF (LES).** EddyFormer using only the LES stream for large-scale dynamics.

- **EF (SGS).** EddyFormer using only the SGS stream for small-scale dynamics.

- **EF (FFT).** EddyFormer variant with SEMConv replaced by Fourier-based convolution. A factorized kernel with the same number of modes as F-FNO is used for a fair comparison.

- **EF (CNN).** EddyFormer variant with SEMConv replaced by standard convolution. A 3D convolution with the same kernel size is directly applied on the collocation points.

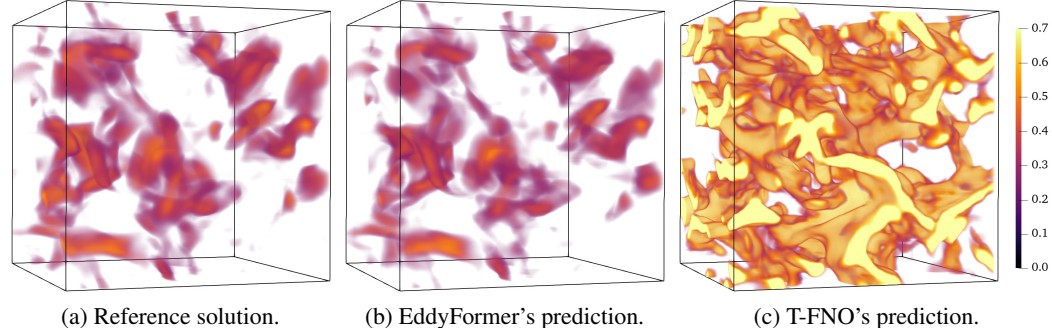

| (a) Reference solution. | (b) EddyFormer's prediction. | (c) T-FNO's prediction. |

Figure 13: Visualization of the density field for a `MHD_64` test sample at $t = 3$. EddyFormer is able to match the reference solution while the baseline T-FNO's prediction diverges after three steps.

Table 13: Direct prediction benchmark on `Re94` : one-step error of different ablation models.

|  | EF | EF (LES) | EF (SGS) | EF (FFT) | EF (CNN) | F-FNO | ResNet |
|---|---|---|---|---|---|---|---|
| Test error | 8.61% | 12.6% | 23.2% | 12.8% | 25.3% | 22.5% | 87.2% |

**Results.**  Tab. 13 reports the one-step error of the ablation models. The results show that the LES stream alone performs worse without the support of the SGS stream, and the SGS stream alone is slightly less accurate than its F-FNO counterpart. This suggests that many global modes in FNOs may be redundant for turbulence modeling. EddyFormer with CNN shows a significant accuracy drop, consistent with the gap between ResNet and FNO. The FFT-based variant of EddyFormer also performs worse, highlighting that the SEM basis offers an advantage over FFT and serves as a core component of our architecture. Compared to the LES-only version, the full model demonstrates that **(a)** SEMAttn captures large-scale dynamics more effectively than F-FNO, and **(2)** SEMConv models small-scale dynamics that FFT-based spectral convolution fail to resolve.

**Choice of the LES filter.**  In classical LES, the filter size defines the cutoff between resolved and subgrid-scale motions, and is typically set based on the grid resolution or an associated spectral cutoff based on the energy spectrum.

Table 14: Data-only benchmark on `Re94` : one-step error across different LES filter sizes.

| $k_{max}$ | 2 | 5 (ours) | 6 | 8 |
|---|---|---|---|---|
| Test error | 18.76% | 8.61% | 8.66% | 8.34% |

To assess the impact of the LES filter size, we vary $k_{max}$ while keeping all other hyperparameters fixed. The results in Tab. 14 show that increasing $k_{max}$ improves accuracy up to $k_{max} = 5$, beyond which the performance saturates. This saturation reflects the limited resolution capacity of the SEMAttn module, which determines the range of large-scale structures the LES stream can effectively capture. When the spectral filter includes higher-frequency modes beyond this capacity, the LES stream is unable to utilize them effectively—especially under a fixed model size. In such cases, increasing the cutoff adds computational cost without improving performance. These results validate our choice of $k_{max} = 5$ as a balance between expressivity and efficiency.

## C.3 The Well dataset

We benchmark EddyFormer on the turbulence-related subsets in The Well. We follow all training protocols described by Ohana et al. [2025], with the exception of the training duration. EddyFormer is trained for 10k steps on each problem, rather than the one-day training budget used in the original benchmark, to ensure reproducibility. The batch size is set to 64. We report the variance-scaled mean square error (VRMSE) scores on test sets as Ohana et al. [2025] did, which is defined as:

$$\text{VRMSE}(u, u') = (\langle |u - u'|^2 \rangle / (\langle |u - \bar{u}|^2 \rangle + \epsilon))^{1/2}, \tag{25}$$

where $u$ is the reference solution and $u'$ is the prediction. The safety value $\epsilon = 10^{-7}$ prevents extreme values on near-constant fields. All reported VRMSE scores are averaged across all time steps in the test set. Baseline results in Tab. 3 are taken from the original benchmark by Ohana et al. [2025].

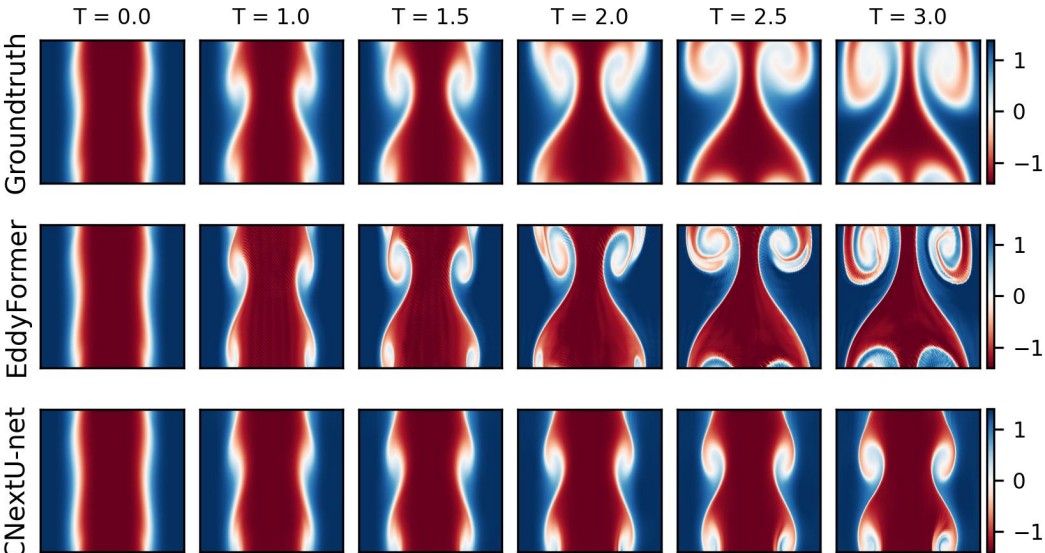

Figure 14: Visualization of the tracer field for a `shear_flow` test sample over time. EddyFormer predicts the development of two vorticies in the shear layer, while the baseline CNextU-net converged to a over-dissipasive solution.

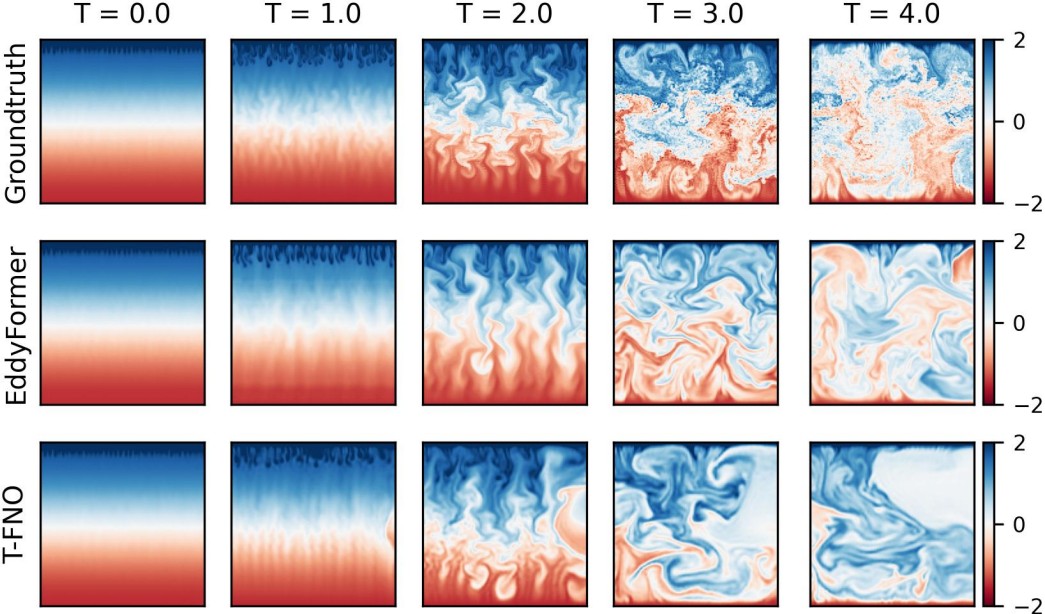

Figure 15: Visualization of the density field for a `rayleigh_benard` test sample over time. Eddy-Former predicts the growth of the mixing layer as well as the plume structures, while the baseline T-FNO predicts a non-physical solution (the solution collapsed into a uniform region at $T = 4.0$).

**Results.** Tab. 15 and Tab. 16 shows the time-averaged per-field VRMSE scores, and Tab. 3 shows the field-average VRMSE scores. Fig. 13, Fig. 14, Fig. 15, and Fig. 16 show visualizations of the `MHD_64`, `shear_flow`, `rayleigh_benard`, and `rayleigh_taylor_instab.` cases, respectively.

Table 15: EddyFormer (Chebyshev basis) on The Well: per-field VRMSE↓ scores.

| | Field | VRMSE |
|---|---|---|
| `MHD_64` | velocity | $\begin{bmatrix} 0.0944 \\ 0.1007 \\ 0.1076 \end{bmatrix}$ |
| | density | 0.1647 |
| | magnetic | $\begin{bmatrix} 0.1368 \\ 0.1052 \\ 0.1028 \end{bmatrix}$ |
| `shear_flow` | velocity | $\begin{bmatrix} 0.0071 \\ 0.3038 \end{bmatrix}$ |
| | pressure | 0.1193 |
| | tracer | 0.0189 |
| `rayleigh_benard` | velocity | $\begin{bmatrix} 0.6221 \\ 0.3112 \end{bmatrix}$ |
| | pressure | 0.0220 |
| | buoyancy | 0.0555 |
| `rayleigh_taylor` | velocity | $\begin{bmatrix} 0.1603 \\ 0.1280 \\ 0.0960 \end{bmatrix}$ |
| | density | 0.0618 |

Table 16: EddyFormer (Legendre basis) on The Well: per-field VRMSE↓ scores.

| | Field | VRMSE |
|---|---|---|
| `MHD_64` | velocity | $\begin{bmatrix} 0.1056 \\ 0.1142 \\ 0.1161 \end{bmatrix}$ |
| | density | 0.1926 |
| | magnetic | $\begin{bmatrix} 0.1611 \\ 0.1219 \\ 0.1161 \end{bmatrix}$ |
| `shear_flow` | velocity | $\begin{bmatrix} 0.0004 \\ 0.1908 \end{bmatrix}$ |
| | pressure | 0.1295 |
| | tracer | 0.0106 |
| `rayleigh_benard` | velocity | $\begin{bmatrix} 0.6289 \\ 0.3063 \end{bmatrix}$ |
| | pressure | 0.0279 |
| | buoyancy | 0.0807 |
| `rayleigh_taylor` | velocity | $\begin{bmatrix} 0.1446 \\ 0.1560 \\ 0.1284 \end{bmatrix}$ |
| | density | 0.0679 |

# D  Limitations

While EddyFormer accelerates the simulation of three-dimensional turbulence, its application to real-world turbulent flows in near-wall regions with fine features is left unexplored. In addition, scaling EddyFormer to accelerate simulations of extreme-scale turbulence at high Reynolds numbers is particularly challenging due to the complex nature of these flows. Future works include extending EddyFormer to general domains and exploiting its scalability.

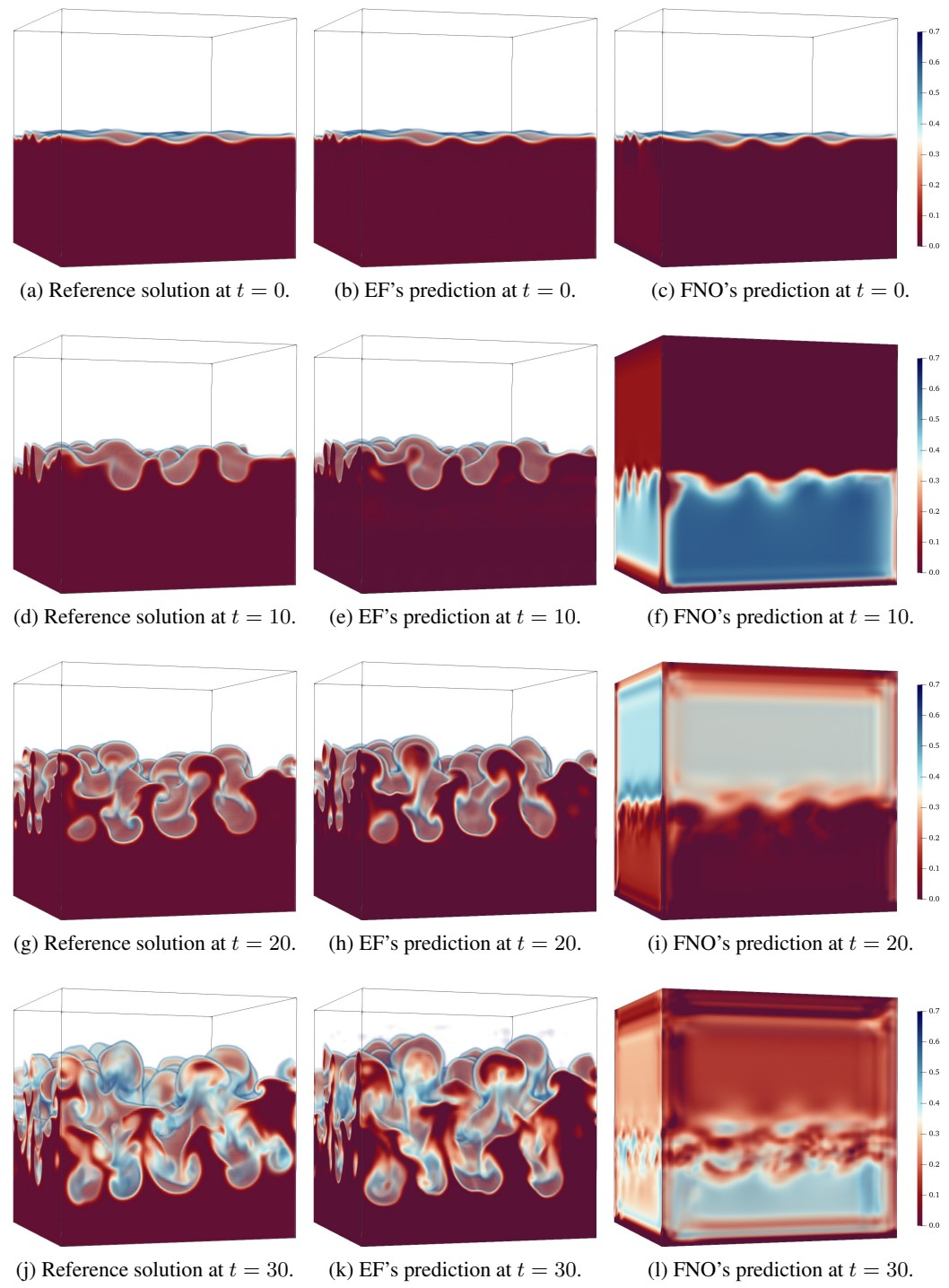

(a) Reference solution at $t = 0$.  (b) EF's prediction at $t = 0$.  (c) FNO's prediction at $t = 0$.

(d) Reference solution at $t = 10$.  (e) EF's prediction at $t = 10$.  (f) FNO's prediction at $t = 10$.

(g) Reference solution at $t = 20$.  (h) EF's prediction at $t = 20$.  (i) FNO's prediction at $t = 20$.

(j) Reference solution at $t = 30$.  (k) EF's prediction at $t = 30$.  (l) FNO's prediction at $t = 30$.

Figure 16: Visualization of the density evolution for a `rayleigh_taylor_instab.` test sample. EddyFormer successfully predicts the development of instability and the plume structures up to $t = 30$, while the baseline FNO fail to converge.

