# OpenReview forum: "EddyFormer: Accelerated Neural Simulations of Three-Dimensional Turbulence at Scale"
_NeurIPS.cc/2025/Conference — NeurIPS 2025 poster_

### Official Review · Reviewer_Qm4N · 2025-06-11

**Clarity:** 2
**Significance:** 2
**Originality:** 3
**Rating:** 3
**Confidence:** 4

**Summary:**

The paper presents a novel method for learned simulation of high-res turbulent flows. It combines a SEM representation, LES-style filtering and a specialized transformer variant. It claims higher accuracy on three experimental domains with 2D and 3D turbulence.

**Questions:**

1. The method needs to be explained better:
  - It sounds like the transformer operates on N^3 tokens, so what happens to the M^3 modes? Are they stacked into the feature dimensions? But also the latent width is specified as 32, so is there a latent encoding from M^3 -> 32 somewhere? What about SEMConv?The paper shouid clarify which operation act in which dimensions, and how those are transformed.
  - If SEMConv also operates on N^3 tokens, why do we want a spectral convolution? The N^3 blocks are in real space, right?
  - SEMConv in general needs a more detailed description (maybe in the appendix)-- padding, integration, etc.
  - How is the model optimized? L2 loss on the output u?
  - Which inputs and outputs are filtered, and how? (3) mentions SGS is unfiltered and LES is filtered, but then 3.1 suggests there are filters W_{in}, and W_{SGS} for SGS?
2. Is there actually any advantage to splitting the model into a SGS, LES path? I understand that the architecture is inspired by LES solvers, but from a model perspective there's little actually enforcing these semantics. Loss is on the sum of both path, and after a layer or two information will be mixed in latents across both branches anyways. Have you tried just using one path (either with both filtered u_SGS and u_LES inputs, or just u) with more layers or higher capacity?
3. Transformer with convolutional blocks is a bit of a weird combination. Can you just use a (bigger) standard transformer with FFNs, maybe with more blocks so that attention can actually do it's thing? Or if the SEMConv layers are very useful, do you need the transformer at all-- i.e. can you just run a FNO-style method on SEM elements?
4. Besides the comments on strange baseline performance numbers, there also isn't any information on how the baselines are parametrized and run. Eg How many layers/width/... (ResNet and FNO can mean a lot of things). What resolution did you choose (you probably can't run a ResNet on 256^3)?  How was the field downsampled?
5. It seems in general FNO performs only a bit worse than EF, but then it completely fails on RB/RT instabilities while EF performs very well. Why do you think that is?

I think this paper really needs a careful revision, investigating baseline performance, and careful examination & ablation of the model architecture. This is probably not feasible to achieve within the rebuttal timeline. While I will increase my score on the off-chance that the authors can quickly clear up the comparisons, and add interesting findings re/ architecture, I'd actually encourage to spend a bit of time on this and resubmit a potentially much stronger & more impactful paper.

**Ethical Concerns:**

["NO or VERY MINOR ethics concerns only"]

**Final Justification:**

After rebuttal and discussion, I still have some reservation about the paper, particularly surrounding baselines, evaluations and missing implementation detail.
The authors have performed some additional experiments and provided some explanations, which will put the paper in a bit of a better spot; I still think a major revisions, with a more thorough set of comparisons and ablations would be best. While I'm still leaning slightly on the negative side I wouldn't object to the paper being published with the additional experiments and exposition. I have hence changed my score from 'reject' to 'borderline reject'.

**Limitations:**

yes.

**Paper Formatting Concerns:**

---

**Quality:**

1

**Strengths And Weaknesses:**

**Strengths**:

3D turbulent flow simulation is quite challenging, and the results are impressive. In general, the paper is well written, provides a nice problem overview & experimental setting. It also performs a very decent evaluations on the experimental results from a CFD perspective, studying e.g. correlation and energy spectra.

**Weaknesses**:

While the paper seems thorough on the CFD side, it falls flat with its contributions to ML.
First, the method description remains vague and at a high level, and it would likely be hard to replicate with the information provided (see questions below for specifics).

Next, the architecture feels quite arbitrary (again see questions for details), and more importantly, the choices aren't ablated at all. So we don't know if any performance gains are due to SEM, or how much the SEMConv layers matter, etc.
And finally, I'm quite concerned about whether baseline comparisons were performed properly. For example, Transolver claims superior performance on several CFD benchmarks, including to FNO-- but here it performs significantly worse than FNO. Worse, KF4 is almost identical to the dataset used in the Kochkov paper, yet in this paper Kochkov seems to perform 10x worse than a simple Resnet-- this can't be right.

It's possible there are certain properties of the domains studied here that cause these differences, or maybe evaluation in these previous works has issues. But this needs to be properly investigated, otherwise we end up with several papers all claiming SOTA with opposing results, and we learn nothing.

Similarly, the impact of the architectural choices needs to be studied and ablated. It's unlikely that someone will reimplement _exactly_ this method as it has a lot of moving pieces--but maybe SEM could be a useful representation for other problems? Or maybe LES splitting, or adding conv layers into a transform is actually very helpful for CFD? These are questions we'd want answered in order for others to build on this work.

---

> ### Author Rebuttal · Authors · 2025-07-31
>
> We thank the reviewer for their detailed feedback. We appreciate the recognition of the difficulty of the 3D turbulence, as well as the acknowledgment of our strong results.
>
> Regarding your concerns, we have performed multiple ablation studies on the LES/SGS splitting and the SEMConv block, as well as introduced two new baseline models to further strengthen our results. We address each of your points below and will incorporate the corresponding updates in our manuscript.
>
> Q1: **Clarifications on how SEMAttn is performed.**
>
> The SEMAttn is defined by Eqn. 11, which mixes features on the same local coordinate from different elements (line 149). Therefore, the $M^3$ modes are neither stacked nor projected, but instead treated independently on every collocation point (line 150).
>
> Q2: **Clarifications on how SEMConv is implemented.**
>
> The SEMConv is implemented by directly evaluating Eqn. 8. Since the input features are also defined on collocation points, the integral can be accurately computed as a weighted dot product using the corresponding Gaussian weights (line 131). Specifically, the kernel is evaluated at each collocation point, and a dot product is taken with the input features, weighted by the quadrature weights. Since our problems have periodic boundary conditions, we do not apply boundary treatment and instead use simple wrapping.
>
> We will revise the manuscript to include these implementation details for clarity.
>
> Q3: **Why do we want a spectral convolution if SEMConv operates “in real space”?**
>
> While SEMConv operates on $M^3$ collocation points (not $N^3$ tokens as you mentioned), spectral parameterization is still preferred over standard convolutions. Spectral models such as FNO have proven highly effective for modeling spatiotemporal dynamics and are significantly more expressive than conventional conv-nets, even though they also operate on real-space inputs.
>
> To validate this, we conducted an ablation study on Re94 by replacing SEMConv with standard convolutions (see table below). Consistent with the performance gap between ResNet and FNO, the modified model exhibits a notable drop in accuracy.
>
> **Table 1**
> | | EF w/ SEMConv | EF w/ CNN | F-FNO | ResNet |
> |---|---|---|---|---|
> | Test error | 8.52% | 25.3% | 22.5% | 87.2%  |
>
> Q4: **How is the model optimized?**
>
> The model is optimized using relative L2 error. (Re94: line 187 and Eq. 14; KF4: line 219 and Eq. 16).
>
> Q5: **Which inputs and outputs are filtered, and how?**
>
> The LES stream uses a spectral cutoff filter with k_max modes for both input and output (lines 115–116). We thank the reviewer for catching the typo on line 119; the correct sentence is: “The initial feature is the projection of the unfiltered input.”
>
> Q6: **The architecture feels quite arbitrary, and the choices aren't ablated at all. So we don't know if any performance gains are due to SEM, or how much the SEMConv layers matter.**
>
> The SEM basis is not just advantageous but essential to our architecture. It enables efficient localized convolutions, and our coarse-mesh attention relies on the locality and interpolation properties of SEM—features not supported by FFT bases.
>
> To assess this design choice, we conducted an ablation on Re94 by replacing SEMConv with an F-FNO layer. The degraded accuracy confirms SEM as a core component. The training speed also shows that SEMConv is more efficient than FFT-based layers.
>
> **Table 2**
> | | EF | EF (FFT) | EF (LES only) | F-FNO |
> |---|---|---|---|---|
> | Test error  | 8.52% | 12.8% | 12.6% | 22.5% |
> | Train speed | 5.2s/it | 6.0s/it  | 2.9s/it | 5.1s/it |
>
> We have already performed an ablation study in Q3 to examine the difference between SEMConv and standard convolution on grid-based inputs, and the results demonstrate that the spectral parameterization in SEMConv is important to model performance.
>
> Q7: **Is there actually any advantage to splitting the model into a SGS, LES path?**
>
> Our core insight is to model large- and small-scale dynamics separately, as they exhibit distinct statistical properties and demand different modeling strategies, as discussed on lines 25–31 and 53–59. A single-stream approach is suboptimal: the LES stream cannot capture fine-scale features, while the SGS stream lacks the global expressivity for large-scale dynamics. Even with inter-scale information transfer, their distinct architectures serve different purposes and are not identical.
>
> To investigate this, we have performed new ablation studies on Re94 using LES/SGS stream only. The results below show that the LES stream on its own is weaker without the SGS stream, and the SGS stream on its own is slightly worse than its corresponding F-FNO. As a side note, this also suggests that many global modes in FNOs might be redundant in turbulence modeling tasks.
>
> **Table 3**
> | | EF | F-FNO | EF (LES-only) | EF (SGS-only) | ResNet |
> |---|---|---|---|---|---|
> | Test error | 8.52% | 22.5% | 12.6% | 23.2% | 87.2% |
>
> To further support the design of our design, we also included TF-Net [2] in our benchmarks (see Table 4 below). TF-Net uses the LES/SGS splitting but still models both streams using standard U-Nets. TF-Net’s original paper already shows that the LES/SGS splitting is beneficial, and here we reproduced their findings. The comparison highlights that our specialized LES and SGS stream designs offer substantial improvements, demonstrating effectiveness of our architecture.
>
> Q8: **Can you just use a (bigger) standard Transformer with more tokens?**
>
> A long-standing issue of using Transformers for spatiotemporal modeling is the unfavorable quadratic scaling of computation w.r.t. the number of tokens. Many previous works took different approaches to mitigate this issue (see discussion in Appendix, line 468-483). For Re94, the total number of modes required to resolve the flow is of $\mathcal O(10^7)$. Therefore, using a bigger transformer is not viable.
>
> To strengthen our claim, we conducted additional benchmarks on Re94 using the Axial Vision Transformer [1] (AViT). AViT employs axial attention on regular meshes, making it a fair and representative Transformer baseline. As shown in the table below, AViT underperforms compared to both FNO and EddyFormer, indicating that pure Transformer architectures are less effective for capturing turbulent dynamics. This result further supports our design choice to integrate Transformers with spectral-based convolution.
>
> **Table 4**
> | | EF    | F-FNO | AViT  | TF-Net |
> |---|---|---|---|---|
> | Test error | 8.52% | 22.5% | 24.5% | 30.4% |
>
> Q9: **I’m concerned about whether baseline comparisons were performed properly. For example, Transolver claims superior performance on several CFD benchmarks, including to FNO -- but here it performs significantly worse than FNO.**
>
> In Tab. 1, we benchmarked several Transformer variants in previous works. They were primarily evaluated on problems with irregular meshes. For instance, Transolver encodes geometric features by grouping similar mesh points into slice-based tokens—a design well-suited for unstructured domains. We believe its stronger performance compared to FNO is due to FNO’s known limitations on irregular meshes.
>
> Additionally, turbulent flows exhibit intricate fine-scale structures that differ substantially from those in prior benchmarks. For example, Transolver was evaluated on FNO’s 2D turbulence dataset, which is relatively less challenging and has already been shown to be well-handled by new models like F-FNO (Fig. 3 in [4]), which we also include as a baseline. In contrast, our 3D turbulence setting presents significantly greater complexity, making it a more demanding benchmark for evaluating model performance.
>
> Q10: **It seems in general FNO performs only a bit worse than EF, but then it completely fails on RB/RT instabilities while EF performs very well. Why do you think that is?**
>
> We note that in our data-only benchmark (Tab. 1), the one-step errors of FNOs are almost 3x larger than EddyFormer’s. This gap is fairly large, and is very likely to be enlarged during the solution rollout (Fig. 8 shows the model prediction after 30 roll-out steps).
>
> RT is unstable w.r.t. perturbations. Error is amplified in the rollout stage, which affects both training and inference stage. We believe EddyFormer performs well due to its slower error accumulation over time, in contrast to FNO, whose errors grow rapidly and lead to instability during rollout.
>
> Q11: **Details on baseline architectures and resolution. How is the reference solution downsampled?**
>
> For F-FNO, we use 64 modes, 4 layers with 32 channels. For FNO, we use 16 modes because of size constraint. For ResNet, we use the standard ResNet-18. For all other Transformers, we use their default architecture.
>
> We refer the reviewer to Tab. 3 for details on dataset resolution, as well as other flow parameters. Re94 is recorded on $96^3$, while KF4 is recorded on $256^2$. The reference solution is downsampled using FFT.
>
> Q12: **KF4 is almost identical to the dataset used in the Kochkov paper, yet in this paper Kochkov seems to perform 10x worse than a simple ResNet -- this can't be right.**
>
> KF4 follows the same problem setting as Kochkov et al., but differs in both dataset and solver. As noted in Fig. 5, their results are based on correcting a $64^2$ finite volume (FVM) solver, which their own paper [3] acknowledges to be less accurate and efficient than spectral methods. We include their result (dashed green line) as a reference only.
>
> In contrast, our results (line 213) are based on correcting a $256^2$ pseudo-spectral solver, which offers much better accuracy and efficiency. For a fair comparison, we also reproduced their setup using a ResNet trained on our high-accuracy dataset by correcting our $256^2$ solver (Fig. 5, solid green line). We will revise the manuscript to clarify this distinction for readers unfamiliar with the background.
>
> ---
>
> [1] arXiv:2310.02994.
>
> [2] arXiv:1911.08655.
>
> [3] arXiv:2207.00556.
>
> [4] arXiv:2111.13802.

---

> > ### Comment · Reviewer_Qm4N · 2025-08-05
> >
> > Thank you for the additional clarifications and experiments-- including those in the paper would definitely strengthen it.
> > While I still feel there's a lot of open questions on which part of the architecture brings most of the value, at least the additional experiments allow us some hints.
> >
> > On Q7, I think there was a misunderstanding. I understand the motivation for the architecture, and why both LES/SGS velocity input is important for the prediction. I was wondering what happens if both LES and SGS velocity, concatenated, are fed to a _single_ prediction branch (probably the LES as it has more capacity). But the findings of the LES-only version are interesting nontheless.
> >
> > The reason I'm asking for these ablations is that neural architectures often don't care that much for our intuitions. Sometimes very simple solutions can outperform more complex ones, even if it goes against our ideas coming from traditional simulation. The most impactful papers are those which can identify such simple, reusable recipes.
> >
> > On the baselines, I'm still not convinced.
> > - Tab.3 only lists dataset, not baseline model details. E.g. if you're using a Resnet-18, do you use its default input size of 224? Or adapt it to 256 as in the dataset? etc.
> > - Q12: In this case I would strongly advise to just remove the Kochkov comparison, this is not an apple-to-apple comparison, and very misleading
> > - Even if the 2D CFD datasets in the Transolver paper are relatively simple, it's still remarkable that it claims superior performance to a host of methods (including FNO) on a large variety of datasets, and here we see the reverse. Similar for the other baseline comparisons-- of course I can't know for sure, but the numbers are very different to other papers and something feels off.
> > - The broader points is, in ML for PDEs we have a crisis where literally every paper claims SOTA, and it's impossible to judge progress. And it is very easy to do-- slightly different datasets, slightly different metrics, and most commonly undertuned baselines. This is hardly the fault of this particular paper, but it doesn't do much to inspire confidence, either. Very little detail on baseline implementation, most experiments are on newly introduced datasets as opposed to established benchmarks, reversal of previous results.
> >
> > It seems the authors made an effort to provide additional experiments, and I don't want to fault a single paper for the failings of an entire subfield. So I will not oppose the paper being published, and will likely raise my score. But I do wish that people would take rigorous comparisons and evaluation more seriously.

---

> > > ### Author Response · Authors · 2025-08-06
> > >
> > > We thank the reviewer for the constructive follow-up. We agree with your broader concerns about evaluation rigor in the ML-for-PDEs community, and expand on this below. We’re grateful that you acknowledge our efforts and are considering raising your score.
> > >
> > > **On Evaluation Rigor**:
> > >
> > > To ensure fair and meaningful comparisons, we evaluate EddyFormer on The Well [1], a well-established benchmark dataset that has been used in prior works. This allows us to directly compare with existing baselines. By grounding our results in a standardized setting, we aim to minimize confounding variables such as dataset shifts or incompatible metrics, and provide a clearer picture of architectural differences.
> > >
> > > We introduced a 3D dataset to address the lack of publicly available benchmarks with comparable difficulty—specifically, those involving high resolution, long rollout horizons, and complex multiscale dynamics. To ensure rigor, we constructed it using carefully calibrated solvers and sufficiently fine discretizations, such that ground-truth errors are negligible compared to learned model error. This avoids a common pitfall where discretization artifacts can obscure true model performance. We have taken evaluation seriously across both 2D and 3D settings, going beyond standard error metrics to include assessments of invariant flow statistics.
> > >
> > > We also confirm that we will release all code, training scripts, configuration files, and datasets used in this paper. This will enable others to reproduce all baseline comparisons and evaluate our method in a transparent and extensible manner. We hope this can contribute to stronger standards in the community and inspire more reproducible comparisons going forward.
> > >
> > > **On Q7**:
> > >
> > > Thank you for the clarification. We now understand that your suggestion refers to feeding both LES and SGS velocities into either a single LES or SGS stream. We note that feeding SGS input into the current LES stream is effectively equivalent to our EF (LES-only) ablation, since the LES path operates on spectrally truncated features—passing high-frequency SGS input through it inherently applies a low-pass filter. On the other hand, our EF (SGS-only) ablation already represents a single-branch model that receives the unfiltered SGS velocity as input. While this does not explicitly concatenate the LES velocity, the LES component is simply a spectrally filtered version of the SGS input. Thus, the convolution layers can in principle recover the LES representation internally, and we expect explicit concatenation to have limited additional effect.
> > >
> > > We also agree that, as of now, there is no clear consensus in the community on which neural architectures generalize best across grid types. As you noted, methods like Transolver may outperform FNO on certain datasets but underperform in others. Our goal was not to claim universal superiority, but to explore where architectural choices like SEM-based design are especially effective for turbulence-related tasks—and to do so in the most transparent way possible.
> > >
> > > **On Baselines and Comparability**:
> > >
> > > All baseline models, including ResNet-18, were adapted to match the input resolution of the dataset, i.e., 256×256 for KF4 and 96x96x96 for Re94. We will clarify this in the appendix.
> > >
> > > We acknowledge that the Kochkov comparison is for reference only. Based on your suggestion, we will remove it entirely to avoid confusion.
> > >
> > > Once again, thank you for your detailed feedback. Your comments have helped us sharpen our contributions, and we fully agree that improving evaluation transparency is a shared responsibility across the community.
> > >
> > > ---
> > >
> > > [1] Ohana, Ruben, et al. "The well: a large-scale collection of diverse physics simulations for machine learning." Advances in Neural Information Processing Systems 37 (2024): 44989-45037.

---

> > > > ### Comment · Reviewer_Qm4N · 2025-08-07
> > > >
> > > > Thank you for your comments, this addresses my questions. I have now updated my review.

---

### Official Review · Reviewer_c37T · 2025-06-28

**Clarity:** 4
**Significance:** 4
**Originality:** 4
**Rating:** 6
**Confidence:** 4

**Summary:**

This work focus on studying neural operators on direct numerical simulation (DNS) dataset. The existing study of neural operators all tends to perform poorly on this challenging task due to the large-scale data (e.g. $256^3$ and $384^3$) and the complexity of the nature of turbulence.

The architecture of the proposed neural operator, EddyFormer, composes of two parts: SEMConv block and SEMAttn block. This design is inspired by the formulation of large eddy simulation (LES), where the solution is decomposed into two parts: LES stream and subgrid-scale (SGS) stream as equation (6). And the two streams are handled by the above blocks accordingly. Namely, SEMConv deals with local eddy dynamics of SGS stream, and SEMAttn handles long-range dependencies of LES stream (See Fig. 2).

Such heuristic design is proven to be highly advantageous with DNS datasets. Datasets are challenging that scale from $96^3$ to $256^3$ and compared with numerical solution on $384^3$ for homogeneous isotropic turbulence. Also, 2D turbulence model, Kolmogrov flow is considered. Finally, turbulence beyond homogeneous isotropic is benchmark in the Well dataset. In three benchmarks, EddyFormer surpasses all baselines by large gap.

**Questions:**

N.A.

**Ethical Concerns:**

["NO or VERY MINOR ethics concerns only"]

**Final Justification:**

While myself was not able to point out further question or issues on this paper, after reading other reviews, I did learn some good points being asked. Some are addressed, and some seem left open. For example,

Addressed:
1. Why choosing convolution-based SEM instead of FFT? The answer is better local feature extraction due to convolution than global basis of FFT.
2. The KF4 dataset is similar to Kochkov paper but different. The scale is $256^2$ here in this paper versus $64^2$. This difference causes the prior art performed much worse on KF4.

Open:
1. The second review from Reviewer inpY mentioned a few prior work in turbulence modeling. It seems the authors did not compare the difference to these works.

Overall, the proposed method is well motivated and highly effective in both accuracy and efficiency. The experiment is conducted comprehensively both on dataset and baselines, showing great advantage. Although the idea of splitting solution as LES may be proposed before with neural networks, this work presents great combination of latest art in neural operators, both convs and transformers, and achieves superior results. Therefore, my score remains to 6.

**Limitations:**

N.A.

**Paper Formatting Concerns:**

N.A.

**Quality:**

4

**Strengths And Weaknesses:**

**Strength**

The proposed method, EddyFormer, is deliberately designed with LES decomposition, which could be the reason for outperforming general neural operators, e.g., FNO and transformer-based neural operators (GNOT, Transolver).

The paper clearly presents the method, and comprehensive experiments are well deployed to show its advantage. DNS is a challenging problem of high significance, and the method is showing strong evidence (in terms of accuracy and speed) of practical applications of neural operators in CFD.

---

> ### Author Rebuttal · Authors · 2025-07-31
>
> We thank the reviewer for their feedback and for recognizing the strengths of our work, including the dual-stream design inspired by LES, and our comprehensive evaluations on turbulence benchmarks. We particularly appreciate the reviewer’s acknowledgment of EddyFormer's improved performance and acceleration on the challenging three-dimensional case.

---

> ### Comment · Reviewer_c37T · 2025-08-06
>
> I think this paper has solid contribution to learning turbulence by neural operators. The main challenges include the large data scale and the highly complex pattern of turbulence. This work achieves both accuracy and efficiency by carefully integrating latest technique of neural operators with the fluid feature. Specifically, the design of local-global architecture matches the LES decomposition which is well-motivated and empirically effective. The experiments are comprehensive with 3 types of challenging  problems and are well conducted, e.g., the resolution is up to $256^3$. All results are showing great advantage of the proposed method.

---

> > ### Author Response · Authors · 2025-08-06
> >
> > We thank the reviewer for their motivating and encouraging feedback. We truly appreciate your positive assessment.
> >
> > During the rebuttal period, we focused on strengthening the paper through new baseline comparisons and ablation studies. We introduced two new baseline models—[TF-Net](https://arxiv.org/abs/1911.08655) and [AViT](https://arxiv.org/abs/2310.02994)—to further support our claims. In addition, we added a direct comparison between EddyFormer and a Fourier convolution-based ablation model, which confirmed the benefit of the SEM basis. We also separately examined the performance of the LES and SGS streams to better understand their individual contributions and validate the design choice of using a dual-branch architecture. These additions helped reinforce the robustness and clarity of our approach.

---

### Official Review · Reviewer_inpY · 2025-07-02

**Clarity:** 1
**Significance:** 3
**Originality:** 2
**Rating:** 4
**Confidence:** 4

**Summary:**

This paper targets deep learning in the context of turbulence simulations, motivating its approach with the well established self similarity of eddies in the turbulence cascade. An architecture is proposed that uses spectral elements in the context of Transformer models, processing inputs in terms of s subgrid-scale and and LES branch. This idea is a classic one, and well-established in turbulence learning [1]. Unfortunately, the paper does not acknowledge this, and instead tries to present this idea as a new contribution, without citing the classic method. While there are clearly differences to [1], the job of the authors would be to discuss and evaluate the differences clearly. Unfortunately, the lack of context and lack of previous work comparisons is continuing later on in the paper.

In the proposed architecture, the SGS part uses FNOs, which are combined with attention based methods for the second branch. The FNO part swaps the FFT for SEM basis functions. The advantages of the SEM approach over an FFT are claimed as a contribution, but its advantages are not made clear in a direct comparison. The FNO approach also means it inherits the negative aspects: especially the bad scaling of FNOs for 3D seems problematic here. Global basis functions for larger 3D resolutions are inherently problematic, which is, however, mitigated with the axial attention.

Scale separation is done via a spectral cutoff, the method uses a manual split into low and high frequencies in each. Low frequencies are provided to the high-frequency branch - in L140 the authors discuss this as surprising, but it's directly in line with the primary scattering of the turbulence cascade. That the other direction is not helpful points to limited accuracy: it's the classic "closure problem" of turbulence, and in the general case the small scales should play a role once the overall approach is sufficiently accurate. (I would recommend to change this paragraph accordingly.)

Due to the lower dimensionality of the low "LES" frequencies, a Transformer with dense attention is applied. The tokens for this stage are likewise processed via the SEM basis functions.

The paper evaluates the method with a learned predictions for 3D isotropic turbulence, and a solver-in-the-loop case for 2D kolmogorov flows. Here, likewise, the authors only focus on the work by Kochkov et al., but do not provide context of their method for newer works from the turbulence area, e.g. [2,3] just to name a few. Generally, the writing and exposition of the paper is problematic - the paper tries to focus on turbulence in its outset, but then seems to turn into a paper focusing on NN architectures.

The authors have also moved related work and the discussion of limitations to the appendix. This naturally frees up room in the main paper, but the NeurIPS page limit is there for a reason. Given the requirements for a NeurIPS paper, these parts should be included in the main text. And naturally, for a paper targeting turbulence, a discussion of ML approaches in this area would be necessary in the main text (not only citing classic text books).

The different methods focus on time stepping, but the paper is unfortunately unclear on the details here. The authors provide details on unrolling (like "N=5"), but don't specify for how many steps the networks were evaluated. Some tests seem to use integer time steps, others like in table 8 use smaller ones. This is crucial information to evaluate the time stability of the method.

[1] Rui Wang, Karthik Kashinath, Mustafa Mustafa, Adrian Albert, and Rose Yu.
Towards physics-informed deep learning for turbulent flow prediction. SIGKDD, 2020
[2] Bjoern List, Liwei Chen, and Nils Thuerey.
Learned Turbulence Modelling with Differentiable Fluid Solvers. JFM, 2022
[3] V. Shankar, D. Chakraborty, V. Viswanathan, and R. Maulik.
Differentiable turbulence: Closure as a partial differential equation constrained optimization. PRF, 2025

**Questions:**

- can the authors demonstrate that the SEM approach has benefits over an FFT by swapping out the basis functions within the full proposed architecture?

- what are the time steps, and how many rollout steps evaluated?

- how is the spectral cutoff chosen, and how much influence does it have. I.e., what is k_max in each case, and how much influence does it have in practice? An ablation would be important here to whether the scale separation actually has a positive influence.

**Ethical Concerns:**

["NO or VERY MINOR ethics concerns only"]

**Final Justification:**

The authors have added comments and clarifications. I think especially the SEM vs FFT comparison is an important one that was missing in the paper.

I think it would be important to double check the final version of the paper, to ensure that the it discusses previous work on learning with scale-separation for turbulence in an appropriate manner. I think it is important for a NeurIPS paper to properly acknowledge widely seen previous work, and this paper clearly reduces the scope of the claims. Also, the authors clearly have work to do to clearly explain their method, the submission had many unclear aspects. Nonetheless, the paper still makes interesting points, so I'm open to supporting an "accept". E.g., the SEM bases are interesting, and scaling up NN-based models to large models above 128^3 is a non-trivial feat.

**Limitations:**

Limitations are not discussed in the main paper.

**Paper Formatting Concerns:**

The paper will need quite some rewriting, but the formatting is fine otherwise.

**Quality:**

2

**Strengths And Weaknesses:**

Strengths

- 3D PDEs , and turbulence are tough and interesting tasks. They're definitely worth studying, and not many papers address 3D up to now.

- the paper shows some strong results: the isotropic turbulence case with 256^3 is neat to see. This is a high resolution, and the "fair" comparison with a high-resolved baseline makes this a strong case

- the paper comes with a good set of "generic" baselines that, however, seem quite arbitrary: the data is structured, then why compare to "unstructured" methods like GNOT and Transolver?

- it's good to see the architecture behaves like a convolutional network and can deal with different domain sizes

Weaknesses

- despite focusing on turbulence, the paper does not provide context of previous work for turbulence-focused ML approaches

- the central SGS/LES split was proposed in previous work many years ago [1]; the authors do not seem to be aware of this; a direct comparison with tf-Net is missing

- the contribution of an SEM approach is likewise claimed as contribution, but it is not mdke clear how much improvements SEM provides over an FFT basis (the Legendre vs Chebyshev comparison implies that the basis does not have a big influence)

- the details on the time discretizations are missing. without this information, the results are difficult to interpret

- similarly, details of the spectral cutoff for the SGS vs LES split are missing

So overall, I think the paper makes some promising steps, but my impression is that it needs a larger, properly checked revision before publication can be recommended.

---

> ### Author Rebuttal · Authors · 2025-07-31
>
> We thank the reviewer for their detailed and constructive feedback. We appreciate the recognition of the novelty and difficulty of the 3D turbulence case, as well as the acknowledgment of our strong results—particularly the high-resolution results and the model’s ability to generalize across domain sizes.
>
> Regarding your concerns, we agree that additional context on prior works, as well as ablation studies on our model architecture, would help clarify our contributions. We have performed additional ablation studies on SEM basis and the LES cutoff frequency, as well as introduced two new baseline models to reflect prior works. We will discuss each point and address them below.
>
> Q1: **Advantages of SEM basis over FFT basis. The Legendre vs Chebyshev comparison implies that the basis does not have a big influence.**
>
> Thank you for raising this important point. We answer and expand on this question here and also add an ablation experiment. The use of the SEM basis is not only an advantage, but a necessary design choice in our architecture. Specifically:
>
> The SEM basis enables efficient implementation of compact convolutions using local spectral elements, which is essential for modeling localized interactions in complex dynamics.
> Our design of self-attention over coarse spatial meshes relies on the locality and interpolation properties of SEM—this would be infeasible using global FFT bases.
>
> The comparison of Chebyshev and Legendre basis does not imply that the SEM basis is an arbitrary choice, but instead shows that multiple bases are compatible in the SEM framework.
>
> On the other hand, if we use FFT basis, the model is limited to global basis, and as you mentioned, exhibits bad scaling w.r.t. the number of modes that it tries to resolve. To assess the impact of this choice, we conducted an ablation study by replacing SEMConv with a standard F-FNO layer, named EF (FFT). The results degraded significantly, highlighting that SEM is advantageous over FFT basis, serving as a core element of our architecture. The training speed results also show that SEMConv with a compact kernel ($s=2$ in this case) is more efficient than FFT on the global field.
>
> Furthermore, we also benchmarked Re94 with an ablation model which only has the LES stream. The resulting accuracy is almost identical to EF (FFT), proving that 1) SEMAttn models large-scale dynamics better than F-FNO does, and 2) SEMConv models small-scale dynamics that FFT basis fails to capture.
>
> **Table 1**
> |             | EF      | EF (FFT) | EF (LES only) | F-FNO   |
> |-------------|---------|----------|---------------|---------|
> | Test error  | 8.52%   | 12.8%    | 12.6%         | 22.5%   |
> | Train speed | 5.2s/it | 6.0s/it  | 2.9s/it       | 5.1s/it |
>
> Q2: **Why compare to "unstructured" methods like GNOT and Transolver?**
>
> We appreciate the reviewer for this important point. GNOT and Transolver are two SOTA Transformers that have been successfully applied to 3D problems. While there are other “structured” models on rectilinear input meshes, e.g., Poseidon and Multiple-Physics Pretraining [1] (MPP), they were only implemented, trained, and evaluated on 2D problems.
>
> To provide an additional comparison, we have adopted AViT [2] (used by MPP [1]) to 3D and benchmarked its accuracy on Re94. We consider AViT a strong and relevant baseline for EddyFormer due to its structured attention mechanism and demonstrated effectiveness on spatially regular data. We set the “patch” size to be $16^3$ for consistency with its 2D counterpart. The results below show that although AViT achieved better performance than other Transformer baselines, its error is still significantly higher than EddyFormer.
>
> **Table 2**
> |            | EF    | AViT  | Transolver | GNOT  |
> |------------|-------|-------|------------|-------|
> | Test error | 8.52% | 24.5% | 57.7%      | 69.7% |
>
> Q3: **A direct comparison with TF-Net is missing.**
>
> We appreciate the reviewer for bringing our attention to this work, and we will discuss it in the paper. As you mentioned, the architecture of TF-Net is fundamentally different from our approach. Most importantly, TF-Net uses three identical conv-nets (at the same resolution) to model the mean and fluctuating components, which do not utilize different properties of these fields.
>
> In FNO’s paper, TF-Net was shown to be outperformed by neural operators. We have also performed additional benchmarking of TF-Net on Re94 to strengthen our argument. The results below show that TF-Net is still worse than F-FNO on three-dimensional problems (F-FNO is best baseline), which is consistent with FNO’s results. In contrast, EddyFormer outperforms both of them by a large margin.
>
> **Table 3**
> |            | EF    | F-FNO | TF-Net |
> |------------|-------|-------|--------|
> | Test error | 8.52% | 22.5% | 30.4%  |
>
> We will revise the manuscript to acknowledge TF-Net, discuss its approach to the LES/SGS split, clarify how our architecture differs, and include this direct comparison.
>
> Q4: **What are the time steps, and how many rollout steps are evaluated?**
>
> The time discretizations for two problems are described on line 178 and line 216, respectively. Re94 uses $\Delta t = 0.5$ and KF4 uses $\Delta t = 1$.
>
> The number of rollout steps are described in Tab. 3 for the training set, and on line 415 for the test set. Re94 is tested for a total trajectory length of $20s$, while KF4 is tested for $50s$. We note that the number of rollout steps is sufficiently large to reach the regime where predictions decorrelate from the ground truth, at which point invariant measures become the more meaningful and reliable indicators of model performance.
>
> Q5: **The details of the spectral cutoff for the SGS vs LES split are missing. How is the spectral cutoff chosen, and how much influence does it have?**
>
> The spectral cutoff frequency $k_\mathsf{max}$, as well as other architecture parameters, are summarized in Tab. 4 and 5. Re94 uses $5^3$ modes and KF4 uses $4^3$ modes in the LES stream.
>
> The value of $k_\mathsf{max}$ is chosen empirically by balancing efficiency and accuracy. The capacity of the SEMAttn determines the range of large-scale structures that the LES stream can effectively capture. If the spectral filter includes excessively high-frequency modes, the LES stream is unable to fully utilize them due to its limited resolution capacity—particularly under a fixed model size. In such cases, increasing the cutoff frequency adds computational cost without improving performance, as the attention module cannot effectively model these finer scales.
>
> To quantify the impact of the spectral filter size, we performed an ablation study on Re94 by varying $k_\mathsf{max}$ while keeping all other hyperparameters fixed. Specifically, we compared models using both smaller and larger filter sizes than our default value of $5$. The results show that model accuracy increases as $k_\mathsf{max}$ grows up to $5$, beyond which the improvements plateau. This saturation indicates that the LES stream reaches its effective resolution limit around $k_\mathsf{max} = 5$, and including higher-frequency modes does not contribute further—validating our choice of spectral cutoff as both sufficient and efficient.
>
> **Table 4**
> |            | EF (cheb, k=2) | EF (cheb, k=5) | EF (cheb, k=6) | EF (cheb, k=8) |
> |------------|----------|----------------------------|----------|----------|
> | Test error | 18.76%   | 8.61%                      | 8.66%    | 8.34%    |
>
> ---
>
> [1] McCabe, Michael, et al. "Multiple physics pretraining for physical surrogate models." arXiv preprint arXiv:2310.02994 (2023).
>
> [2] Ho, Jonathan, et al. "Axial attention in multidimensional transformers." arXiv preprint arXiv:1912.12180 (2019).

---

> > ### Comment · Reviewer_inpY · 2025-08-06
> >
> > I want to thank the authors for the additional comments and clarifications. I think especially the SEM vs FFT comparison is an important one that was missing in the paper.
> >
> > I think it would be important to double check the final version of the paper, to ensure that the it discusses previous work on learning with scale-separation for turbulence in an appropriate manner. Nonetheless, I'm open to raising my score pending on the post-rebuttal discussion.

---

> > > ### Author Response · Authors · 2025-08-06
> > >
> > > Thank you for your thoughtful feedback and for considering raising your score. We’re glad to hear that our additional experiments help strengthen the overall contribution.
> > >
> > > We appreciate your point regarding prior work on scale separation in turbulence modeling. We will carefully revise and update the final version of the paper to ensure that the discussion of related work, particularly in the context of multiscale learning, is accurate, comprehensive, and well-situated within the broader literature. In addition, we will take this opportunity to further clarify our core contributions, and provide more detail on the model architecture and accompanying ablation studies, as requested by other reviewers.

---

### Official Review · Reviewer_pTWs · 2025-07-20

**Clarity:** 3
**Significance:** 3
**Originality:** 2
**Rating:** 4
**Confidence:** 4

**Summary:**

This paper proposes EddyFormer for ML-based modeling of turbulence. The main idea is to take inspiration from large-eddy simulations and use two sub-networks: standard-attention for modeling large-scale and spectral convolutions for modeling subgrid-scale turbulence. EddyFormer is benchmarked on different turbulence problems from the Well dataset. Results show that EddyFormer outperforms neural operator (FNO, FFNO, TFNO variants) and transformer (GNOT, Transolver) baselines. The paper also reports 30x speedup over DNS simulations using efficient pseudo-spectrak solvers on a single GPU.

**Questions:**

Please see weaknesses listed above.

**Ethical Concerns:**

["NO or VERY MINOR ethics concerns only"]

**Final Justification:**

The authors present results for total energy over 50 time steps and claim they preserve energy balance compared to F-FNO but the numbers go down, then up, then down so the conclusion of significant systematic dissipation is unconvincing. Scalability to domains beyond 256^3 is still an open question that's unaddressed. For these reasons, I keep my current score.

**Limitations:**

Data efficiency is not addressed. There is an underlying assumption that there is access to large training data for transformer models but this is not the case in turbulence (specifically DNS data as used in this paper).

**Quality:**

3

**Strengths And Weaknesses:**

Strengths:

-- Builds on turbulence theory and well-established modeling methodologies (i.e., LES) motivating the dual-stream design underlying EddyFormer.

-- Experimental evaluations consider challenging benchmarks such as Rayleigh-Taylor instability and demonstrates improved performance compared to baselines.

Weaknesses:

-- The evaluation of long-time rollout is limited in duration. If delta t = 1 s, the maximum number of time steps in the examples is atmost 50.

-- The design lacks any specific treatment for flows with wall effects and is unclear how the current architecture will generalize to real-world turbulence with irregular geometries or wall-bounded flows.

-- Evaluation metrics are limited to relative L2 and doesn't check if physical quantities are conserved. Since the model also doesn't enforce any hard physical constraints (e.g., conservation laws), it's unclear if conservation is met and how these models will generalize to OOD.

-- The reported speedup is for a specific problem size (256^3) for a specific benchmark. It is unclear how EddyFormer scales at larger sizes and how that compares to DNS scaling. Quadratic scaling in attention costs might limit the scalability of the model at higher resolutions.

---

> ### Author Rebuttal · Authors · 2025-07-31
>
> We thank the reviewer for their thoughtful feedback and for recognizing the strengths of our work, including our integration of turbulence theory, the dual-stream design inspired by LES, and our comprehensive evaluations on challenging turbulence benchmarks. We particularly appreciate the reviewer's acknowledgment of the improved performance demonstrated by EddyFormer against strong baselines.
>
> We have performed additional energy conservation checks, as well as introduced a new benchmark on domains with irregular geometries. Regarding your concerns on our evaluation and the applicability of EddyFormer, we address each point below.
>
> Q1: **Evaluation metrics are limited to relative L2 and don't check if physical quantities are conserved. Since the model also doesn't enforce any hard physical constraints (e.g., conservation laws), it's unclear if conservation is met and how these models will generalize to OOD.**
>
> We acknowledge the importance of evaluating conservative physical quantities. We want to highlight that we have already conducted extensive evaluations that go beyond standard error metrics (like L2 relative error) to assess whether invariant physical properties are preserved implicitly. Specifically, we have:
>
> - **Energy spectrum**. In Fig. 7, we compare the energy spectrum of the model predictions with ground truth, capturing how well the model preserves energy across different scales.
> - **Structure function**. In Fig. 3c and 5c, we evaluate the two-point third-order structure function, a key diagnostic in turbulence that reflects inter-scale energy transfer. As discussed around Eq. 18, this metric directly tests whether the model preserves correct inter-scale statistics, which are tightly linked to conservation laws in turbulent systems.
> - Furthermore, since the turbulent flow cases are in statistical equilibrium, evaluating model predictions over long temporal rollouts already implicitly tests for conservation-like invariants. The sustained match over time between predicted and reference statistics indicates that the model generalizes well and does not accumulate unphysical errors.
>
> Nevertheless, we have performed additional energy conservation checks on KF4 and find that EddyFormer is able to preserve energy balance over an extended period of time. This flow problem receives energy injection from the sinusoidal forcing, and dissipates energy at the smallest scales. Specifically, the energy components are:
>
> \begin{equation}
>   E(t) = \langle \frac 1 2 \mathbf{v}(t, x)^2 \rangle_x, \quad
>   \frac {dE_\mathsf{dis}} {dt} = \nu \langle \nabla \mathbf{v}(t, x)^2 \rangle_x, \quad
>   \frac {dE_\mathsf{inj}} {dt} = \langle \mathbf{f}(x) \cdot \mathbf{v}(t, x) \rangle_x,
> \end{equation}
>
> To test energy conservation, we measure all three components of the energy budget, and record the accumulated energy $E_\mathsf{tot}$, $E_\mathsf{tot}(t) = E(t) + \int_0^t dE_\mathsf{dis} - dE_\mathsf{inj}$, over 50s on the test set. The results below show that EddyFormer is able to preserve the energy balance over an extended period, while F-FNO (the model that we find is second best for this problem, among our benchmarks) shows a significant and systematic dissipation over time.
>
> **Table 1**
> | $E_\mathsf{tot}(t)$ | 0~5 | 5~10 | 10~15 | 15~20 | 20~25 | 25~30 | 30~35 | 35~40 | 40~45 | 45~50 |
> | --- | --- | --- | --- | --- | --- | --- | --- | --- | --- | --- |
> | EddyFormer | 1.728 | 1.668 | 1.692 | 1.660 | 1.712 | 1.736 | 1.695 | 1.694 | 1.706 | 1.670 |
> | F-FNO | 1.728 | 1.668 | 1.688 | 1.588 | 1.650 | 1.623 | 1.524 | 1.518 | 1.502 | 1.511 |
>
> Q2: **The evaluation of long-time rollout is limited in duration. If delta t = 1 s, the maximum number of time steps in the examples is at most 50.**
>
> Thank you for your comment. We note that our rollout of 50s in KF4 already represents a significantly longer duration than many related works. For example:
>
> FNO reports rollouts of up to 30s,
> PDE-Refiner adopts a similar temporal range of 15s,
> APEBench [2], which specifically targets long-rollout evaluation, uses 40 steps at most.
>
> Thus, our rollout length is among the longest in current literature and sufficiently captures the chaotic regime where prediction stability and invariant measures are most challenging.
>
> Q3: **The design lacks any specific treatment for flows with wall effects and is unclear how the current architecture will generalize to real-world turbulence with irregular geometries or wall-bounded flows.**
>
> This is an important consideration. The SEM framework underlying EddyFormer inherently supports geometric flexibility, making it naturally compatible with wall-bounded flows and irregular geometries. Although our current study primarily focuses on turbulence in homogeneous domains, extending EddyFormer to incorporate explicit wall effects and complex geometries represents a promising direction for future work, leveraging this intrinsic compatibility. We will discuss this point and make it more clear in our updated manuscript.
>
> To test this, we have also performed an additional experiment on acoustic scattering in a maze. This is another subset in The Well, featuring wave reflections in complex geometries. We also follow the training protocol described in The Well, and report the VRMSE scores below.
>
> **Table 2**
> |            | FNO    | TFNO   | U-Net  | CNextU-Net | EF (leg) |
> |------------|--------|--------|--------|------------|----------|
> | Test VRMSE | 0.5062 | 0.5057 | 0.0351 | 0.0153     | 0.01932  |
>
> Due to time constraints, we didn’t modify the model to support wall treatment using attention masking, etc. Still, EddyFormer is able to match the performance of the best baseline. The comparisons with FNOs show that the geometric flexibility of SEM is significant.
>
> Q4: **The reported speedup is for a specific problem size (256^3) for a specific benchmark. It is unclear how EddyFormer scales at larger sizes and how that compares to DNS scaling. Quadratic scaling in attention costs might limit the scalability of the model at higher resolutions.**
>
> We acknowledge the importance of evaluating and extending EddyFormer to larger problem sizes. While our current implementation is tested on $256^3$ resolution, EddyFormer adopts standard attention mechanisms similar to those used in large language models, enabling the use of well-established techniques (such as memory-efficient attention variants and system-level optimizations) to scale to significantly larger contexts. For extreme-scale turbulence problems, additional architectural adaptations, such as localized attention windows or hierarchical representations during training, can further improve scalability. We view this as a promising and tractable direction for future work.
>
> Q5: **Data efficiency is not addressed. There is an underlying assumption that there is access to large training data for transformer models but this is not the case in turbulence.**
>
> Thank you for this point, we will make the data efficiency aspect more clear, and expand on it here: we note that we use ~100k snapshots to train both KF4 and Re94. For previous works, both FNO and JAX-CFD also use ~100k solution snapshots. We highlight that Re94 is a three-dimensional problem, which exhibits more complex dynamics. Yet, EddyFormer is able to achieve high accuracy and generalization within the 100k data limit.
>
> ---
>
> [1] Lippe, Phillip, et al. "Pde-refiner: Achieving accurate long rollouts with neural pde solvers." Advances in Neural Information Processing Systems 36 (2023): 67398-67433.
>
> [2] Koehler, Felix, Simon Niedermayr, and Nils Thuerey. "APEBench: A benchmark for autoregressive neural emulators of PDEs." Advances in Neural Information Processing Systems 37 (2024): 120252-120310.

---

> > ### Comment · Reviewer_pTWs · 2025-08-09
> >
> > Thank you for your responses.

---

### Note · Authors · 2025-08-15

We thank all reviewers for their time and constructive feedback throughout the review process.

In this work, we introduced a new architecture based on large eddy simulations (LES), designed to tackle challenging multiscale dynamics in turbulent flows. Through careful experimentation, we demonstrate that our spectral element methods (SEM)-based design enables efficient and accurate modeling in regimes that are difficult for existing methods—delivering strong results on both standard benchmarks (such as The Well) and newly proposed high-resolution 3D datasets. Our evaluations go beyond L2 error metrics to include various invariant statistics of fluid flow, which can be more holistic and meaningful for fluid dynamics simulations.

We particularly appreciate the thoughtful suggestions on architectural ablations, baseline comparisons, and evaluation practices. These comments have significantly strengthened the clarity and rigor of our work. In response, we conducted additional experiments during the rebuttal period, including:
- A direct comparison between our SEM-based model and a Fourier convolution-based variant, which confirmed the necessity of the SEM basis for efficient multiscale modeling;
- Ablations that separately examine the roles of the LES and subgrid scale (SGS) streams, validating the motivation behind the dual-branch architecture;
- The addition of two new baseline models (TF-Net and AViT) to further support the strong performance of our model.

We also acknowledge the importance of situating our contributions within the broader context of multiscale learning for turbulence, and we will revise the final version of the paper to more carefully discuss prior work on scale separation. In addition, we will further clarify our core contributions, architectural choices, and evaluation methodology in the camera-ready version, including details regarding baseline implementation to ensure fair and transparent comparisons.

We plan to release all code, trained models, and datasets—including our new 3D dataset, which was calibrated using fine discretizations, and is intended to support challenging benchmarking on high-resolution, long-horizon, multiscale turbulent flows.

We appreciate all reviewers’ engagement, and we hope our revisions and clarifications have addressed your concerns. Thank you for your consideration.

---

### Decision · Program_Chairs · 2025-09-17

**Decision:**

Accept (poster)

**Comment:**

This paper presents a novel neural operator architecture that combines spectral element convolutions and transformer-based attention to efficiently model large-scale three-dimensional turbulence. Key idea is model large-scale (grid-scale) and small-scale (subgrid-scale) dynamics separately. The initial reviews for this work were highly divergent, spanning the full range. There were significant initial weaknesses in its framing, methodological rigor, and connections within prior work. The discussion and author rebuttal have been pivotal in clarifying the paper's standing. Further validation on more diverse and practical turbulence scenarios would strengthen the case for broader impact.

Despite some lingering reservations from the reviewers I recommend acceptance as the flaws are not fatal. Honestly, some of it is a critique of community-wide practices, and the authors have made a good-faith effort to improve transparency.